# QAProt: Enabling Sequence-to-Text Protein Function Learning with a Comprehensive QA Corpus

## Abstract

Inferring protein function from sequence is a grand challenge in genomics, yet progress is bottlenecked by the narrow, template-driven datasets available for training. These datasets, derived from structured databases, fail to leverage the rich diversity of knowledge in scientific literature. To address this gap, we introduce **QAProt**, a large-scale corpus with over 987,000 free-form question-answer pairs mined directly from PubMed abstracts, capturing broader topical and linguistic variability than existing resources. To ensure high fidelity, we developed a rigorous multi-LLM cleaning pipeline that yields a $\sim 13\times$ reduction in estimated hallucination rates. Our analyses reveal that current protein LLMs exhibit a performance collapse when tested on the realistic distribution of taxa and functions found in QAProt, highlighting the complementary nature of our literature-derived data distribution. A single epoch of fine-tuning on our dataset yields remarkable improvements, including an **86% performance gain** on previously unseen protein domains. QAProt is a complementary new resource that enables the development of more powerful, generalizable models for protein science. Dataset available anonymously at https://huggingface.co/conferenceacc/QAProt.

## 1 Introduction

Proteins are the molecular machinery that underpins nearly every cellular function, whose function depends on their three-dimensional structure. The three-dimensional structure of a protein can now be directly predicted from its amino acid sequence by models such as AlphaFold (John Jumper & Hassabis, 2021), ESMFold (Lin et al., 2022a), and RosettaFold (Baek et al., 2021). Although structure prediction accelerates structure-based drug discovery and protein engineering, translating these structural insights into clear descriptions of biological function and in vivo behavior remains a difficult and open-ended challenge. Bridging this gap demands sequence-to-text translation models. That is, models that can first recognize patterns (such as domains, functional residues, motifs) in amino acid sequences, and then learn to map those sequence patterns to the many biochemical roles, interactions, and phenotypic effects that define protein behavior.

To enable such sequence-to-text translation, several protein question–answer benchmarks have been recently developed. Several approaches used structured Knowledge bases to generate QA. For example, UniProtQA (Luo et al., 2023) generated a template question from structured UniProt annotation fields. Similarly, PQA uses expert-defined prompts to convert SwissProt annotations into questions and answers via GPT-3.5, while applying sequence-clustering to debias large protein families and reporting minimal hallucination in human evaluations (Carrami & Sharifzadeh, 2024). In contrast, vision-language research (e.g., LLaVA (Liu et al., 2023a)) has shown that mining unstructured captions can yield rich, free-form VQA data—yet no comparable unstructured approach has been applied to the protein domain. These structured methods have laid important groundwork by formalizing protein function as a natural language task; however, we show

here their dependence on fixed templates and database fields constrains the diversity of question styles and biological themes they can cover.

To address this gap, we introduce QAProt, a diverse open-ended protein QA dataset developed to date. By mining scientific literature published through 2020, we extracted over 987,000 free-form question–answer pairs covering structural, functional, evolutionary, and biomedical aspects of proteins. This literature-based approach significantly enhances the diversity of protein QA compared to existing datasets, introducing a broader range of query types—from straightforward domain identification to intricate mechanistic and contextual questions. As shown in Appendix Table 7, QAProt is a free-form question format and openly available data that relies on scientific abstracts from PubMed. QAProt spans proteins, with moderately long questions and answers, longer average protein sequences (Table 1). Our data collection process involves three steps: scientific abstract collection, QA extraction using LLMs, and data cleaning and refinement. Consequently, QAProt provides a rich and varied resource for training language models, facilitating the translation of amino acid sequences into detailed, human-readable functional annotations.

## 2 RELATED WORK

**Structured Protein QA Datasets.** Datasets such as UniProtQA (Luo et al., 2023), PQA (Carrami & Sharifzadeh, 2024), ProtDescribe (Xu et al., 2023), and Mol-Instructions (Fang et al., 2024) predominantly utilize structured annotations derived from UniProt Swiss-Prot (Consortium, 2022). UniProtQA and ProtDescribe use a fixed subset of manually curated fields (e.g., function, name, family, and subcellular location), resulting in templated questions and descriptions prone to biases from overrepresented annotations and protein families. Similarly, PQA leverages structured annotations coupled with GPT-3.5 prompting, employing a 50% similarity threshold (Consortium, 2022; Coudert et al., 2022) to mitigate biases and training-test leakage. Mol-Instructions, although concurrently introduced, extends annotations to various biomolecular tasks, but remains primarily Swiss-Prot-curated for its protein-specific subset. Collectively, these structured datasets exhibit inherent biases toward well-characterized proteins from heavily studied organisms, suffer annotation inconsistencies, and—as we demonstrate—lack diverse question types. Moreover, annotations from Swiss-Prot often rely on homology rather than experimental validation, introducing noise and uncertainty.

**Protein-related LLMs.** Recent approaches applying multimodal LLMs to proteins frequently leverage these datasets. Techniques inspired by vision-language alignment (Alayrac et al., 2022; Liu et al., 2023a) integrate protein sequences with natural language text, treating proteins as a distinct modality (Huo et al., 2024; Wang et al., 2024; Liu et al., 2024; Luo et al., 2023; Fang et al., 2024; Xiao et al., 2024). For instance, BioMedGPT combines UniProt-QA with PubChem-QA derived from PubChem to address complex biomedical queries, while Galactica (Taylor et al., 2022) uses general scientific corpora for broader reasoning tasks, including protein captioning. Beyond amino acid sequences, studies have integrated multimodal LLMs with DNA/RNA sequences (Richard et al., 2024) and protein 3D structures (Huo et al., 2024; Wang et al., 2024). A detailed survey of protein language models, instruction tuning, and broader multimodal LLM developments is available in the Appendix Section A.

## 3 QAPROT DATASET

QAProt is a question-answer dataset that contains 987,588 questions and answers. The dataset information includes the UniProt accession ID, protein name, organism name, and amino acid sequence. The data in QAProt is sourced from PubMed, a biomedical literature database. Table 1 summarizes the total number of QA pairs and other statistics across different proteins.

The construction of QAProt involves four main stages: data preprocessing, LLM-based extraction, named entity recognition, and data cleaning and validation. Figure 1 illustrates the overall pipeline. We sampled

| Dataset | Number of QA Pairs | Avg. Length | | |
|---|---|---|---|---|
| | | Questions | Answers | Sequences |
| ProteinChat | 1,154,979 | 7.86 | 1.72 | 363.81 |
| UniProtQA | 1,513,126 | 8.30 | 14.35 | 361.84 |
| PQA (Pika) | 1,705,357 | 8.09 | 10.59 | 409.06 |
| ProteinKG25 QA | 3,034,098 | 6.87 | 7.05 | 337.70 |
| Swiss-Prot QA | 1,224,658 | 1.22 | 13.36 | 335.88 |
| PDB-QA | 4,199,775 | 8.82 | 1.19 | 290.81 |
| Mol-Instructions | 495,004 | 16.59 | 27.86 | 420.41 |
| **QAProt (Ours)** | 987,588 | 8.76 | 12.52 | 552.54 |

Table 1: Main statistics of the datasets used for the experiments in the study. For the average length section, questions and answers are measured in words, while sequences are measured in characters.

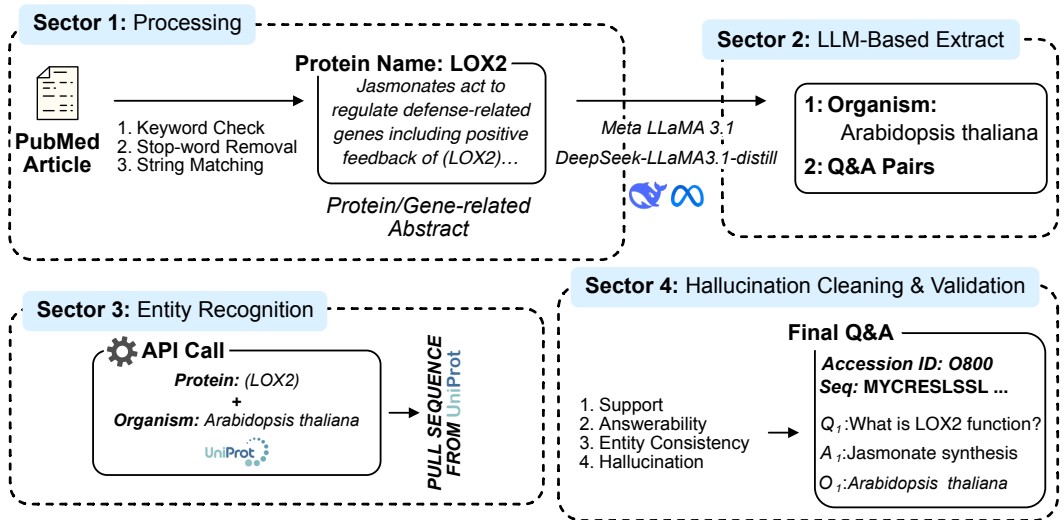

Figure 1: **A high overview of QAProt pipeline.**

140 instances from QAProt and analyzed the question types, with the results summarized in Table 2. The questions are diverse, covering protein function and structure, interactions, localization and expression, disease associations, as well as comparative relationships. In addition, QAProt is designed to align sequence-level information with textual knowledge by providing explicit molecular entities and assigning a PubMed ID to each instance, which ensures that QA pairs remain closely linked to their original literature context. These features make QAProt an ideal resource for training and evaluating biomedical large language models (LLMs).

## 3.1 DATA PREPROCESSING

Using the NCBI Entrez API, we obtained 250,000 structured article records from the NCBI PubMed database. These records include the PubMed ID, article title, abstract text, and other metadata (see Section B.2 for details). For this study, we built our primary dataset from abstracts due to two key scalability challenges: (1)

limited public access to many full texts, and (2) the high computational cost of processing longer documents at scale. More detailed discussion and findings on using full articles can be found in Section 4.6 and Section 5.

Next, we retain abstracts that specifically discuss proteins and genes using UniProt API. Given an abstract, we iterate over the words in the abstract, and for every word, we make an API call to UniProt to check whether the word is a protein name or not. We excluded the stop words and performed string matching techniques to remove unnecessary words/info, such as URLs. If an abstract returns at least one protein/gene name, that abstract is retained; otherwise, it is excluded. The

Sequence: MCDFTEDQ...

| Question Type | Example Questions |
|---|---|
| Function | What is the *function* of the protein? |
| Localization | *Where* is the protein localized in the cell? |
| Expression | In which *tissues* is the protein expressed? |
| Disease | *Mutations* in the protein is associated with what disease? |

Table 2: Question types used in our sequence-conditioned QA task. Given the protein sequence shown above, the model input is `<Sequence> <Question>` and the output is a free-text `<Answer>`.

final number of retained abstracts is 182,975. This step is only to filter abstracts, while the correct protein and organism names mentioned in the abstract will be extracted more strictly, as discussed in later sections.

### 3.2 LLM-BASED EXTRACTION

**QA Extraction.** Next, we extracted implicit questions from abstracts that pertain to protein attributes (e.g., function, structural motifs). For QA extraction, we employed the DeepSeek distilled version of LLaMA3.1 to generate free-form QA pairs from PubMed abstracts. The LLM was tasked with understanding the abstract content, directly identifying protein names, and embedding them into the generated QAs. Figure 5 illustrates the overall pipeline, including the prompt, sample abstract, and generated questions and answers. To scale processing, we executed multiple parallel jobs across the cluster, totaling approximately 3200 GPUs (400 nodes) on the Frontier supercomputer at the Oak Ridge Leadership Computing Facility. After this step, every PubMed ID will be associated with the mentioned protein names and a list of questions and answers that discuss the proteins based on the information from the abstract.

**Organism Extraction.** The function, structure, and interactions of a protein often depend on its species context. Although the same protein name may appear across different species, they do not represent identical molecules. Therefore, extracting the amino acid sequence requires specifying the corresponding species version. So we perform another LLM extraction using the DeepSeek-r1 model to obtain the main organism discussed in the abstracts. Figure 6 demonstrates the prompt used for this task and an example input–output from our dataset. The LLM is instructed to return two kinds of output: an actual organism (e.g., "Homo sapiens") or "Unknown" if the organism is unclear from the text. Abstracts where the LLM returns "Unknown" are discarded.

### 3.3 PROTEIN NAMED ENTITY RECOGNITION (NER) MODULE

Based on the protein name obtained from UniProt and the organism identified in the previous step, we used the UniProt API (Swiss-Prot section) to map each protein name in the QA pairs to a standardized accession ID. In this way, we uniquely determined a protein entry and eliminated ambiguities caused by identical protein names appearing across different species.(Consortium, 2022). Specifically, we queried protein and organism names through the UniProt Query API, retrieving accession IDs, standardized names, and organism details from Swiss-Prot. The first entry of the query output matching any input keyword was considered a valid match. This mapping yielded approximately 987,588 QA pairs across 125,000 abstracts (Table 1)

Once the Uniprot ID is known, amino acid sequences were retrieved using the UniProt Query API via the `{uniprotkb/accession.fasta}` endpoint. Duplicate accession IDs were cached to reduce network

overhead. Missing sequences (e.g., due to incomplete records) were labeled "Not found" and excluded. All retrieved sequences were validated against expected Swiss-Prot length ranges, resulting in ∼3,356,001 unique accession IDs. Overall, this process not only standardized diverse organism synonyms by mapping them to unique UniProt entities but also enriched textual data with manually curated protein annotations and amino acid sequences, thereby providing biologically meaningful context for QA pairs.

### 3.4 DATA CLEANING AND VALIDATION

In open-domain protein question answering (protein QA), large language models (LLMs) may generate answers inconsistent with the source literature (so-called "hallucinations"), which undermines dataset reliability and the trustworthiness of downstream tasks. To address this, we constructed a rigorous cleaning protocol and deliberately selected three independent and diverse LLMs (`meta-llama/llama-3.3-70b-instruct`, `x-ai/grok-3-mini`, `deepseek/deepseek-chat`) to avoid self-assessment bias. Each QA pair was fact-checked and entity-verified against its source PubMed abstract, yielding structured quality control flags (Pass/Fail with reasons). Only QAs unanimously judgMed as "Pass" were retained.

The cleaning process specifically checked the following three criteria:

1. **Support**: whether the answer is fully supported by the abstract.

2. **Answerability**: whether the question can be answered solely from the abstract.

3. **Entity Consistency**: whether all proteins, genes, organisms, and other biological entities match the abstract verbatim or semantically.

This process results in two versions of the dataset: the Original QA set and the newly created Cleaned QA set.

## 4 EXPERIMENTS & RESULTS

We evaluate QAProt systematically across key dimensions of its utility as an open-ended protein QA dataset: hallucination estimation, semantic topic coverage, linguistic diversity, and quantitative benchmarking.

### 4.1 HALLUCINATION RATE EVALUATOR

To provide an unbiased assessment of our cleaning protocol, we employed three different, state-of-the-art Hallucination Evaluator LLMs (Claude Sonnet 4, Gemini 2.5 Pro, and GPT-4o). We choose these separate sets of models for evaluation to avoid any self-assessment bias. We randomly selected 2,000 PubMed abstracts and their corresponding QA pairs from each dataset, the Original QA and Cleaned QA datasets. The Hallucination Evaluators independently assessed these samples against the same three criteria mentioned above. A QA pair was flagged as a "hallucination" if it failed to meet one or more criteria. The results, presented below, show the hallucination rates determined by each reviewer. As shown in Table 3, this analysis reveals a substantial improvement; compared to the original dataset, the cleaned dataset achieved an average **13-fold reduction** in hallucination rate. The hallucination rate is defined as the number of answers that are not supported by the source abstract, divided by the total number of answers that require verification.

### 4.2 COMPARISON OF DATASET DIVERSITY

To analyze semantic, topical, and linguistic diversity, we conducted a comparative study of QAProt against three existing protein QA datasets: **ProteinChat**, **Mol-Instructions**, and **PQA (Pika)**. The results show that **QAProt** clearly outperforms the others in both semantic/topic coverage and lexical richness(Figure 2).

| Evaluator | Original QA (%) | Cleaned QA (%) | Reduction Factor |
|---|---|---|---|
| Claude Sonnet 4 | 20.58 | 1.91 | 10.8× |
| Gemini 2.5 Pro | 17.77 | 2.77 | 6.4× |
| GPT-4o | 14.07 | 0.61 | 23.1× |
| **Average** | 17.47 | 1.76 | ~13× |

Table 3: Hallucination rate comparison between Original QA and Cleaned QA datasets, evaluated by three independent LLMs.

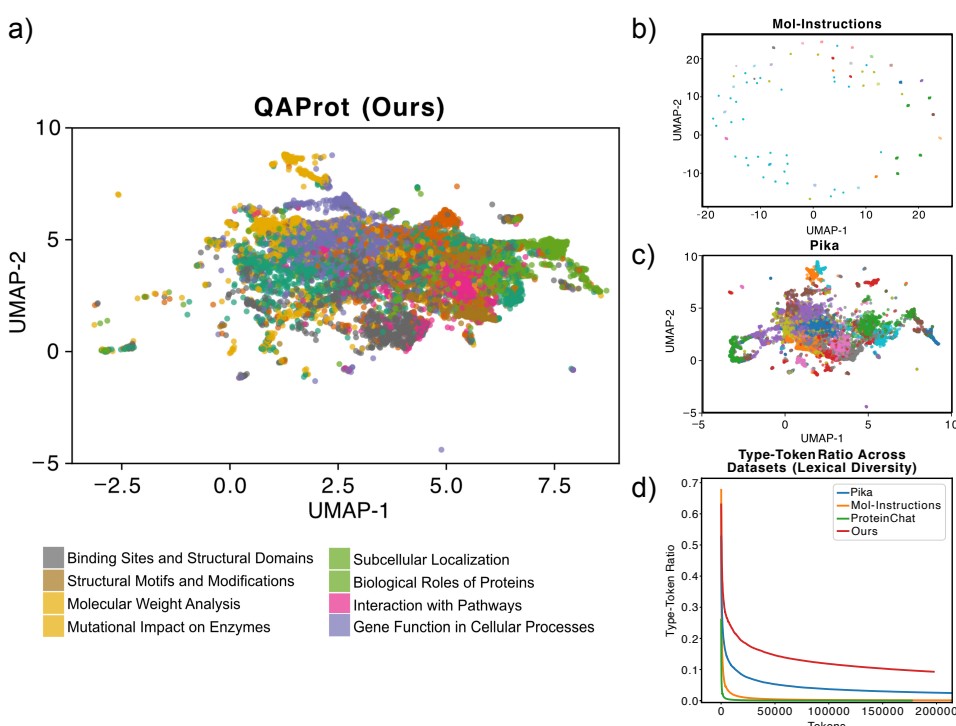

Figure 2: **Data diversity across protein QA datasets**. (a–c) UMAP of BioMedBERT question embeddings, colored by inferred topic: (a) QAProt; (b) Mol-Instruction; (c) PQA. QAProt 22 well-separated, evenly sized clusters spanning diverse biological themes. (d) **Lexical diversity across protein QA datasets**: Type–Token Ratio vs. tokens: QAProt (red) sustains high lexical diversity, whereas PQA (blue), ProteinChat (green), and Mol-Instructions (orange) rapidly converge toward zero, reflecting heavy template reuse.

At the semantic level, clustering and UMAP visualization based on **BioMedBERT** embeddings indicate that QAProt spans around 22 topical clusters, far exceeding the more templated UniProt-derived datasets. Moreover, QAProt includes themes absent from these structured sets, highlighting the complementary nature of PubMed-derived unstructured QA compared to UniProt-based QA.

At the linguistic level, **TTR** analysis shows that ProteinChat and Mol-Instructions suffer from extreme redundancy, while PQA performs slightly better but remains limited. In contrast, QAProt consistently maintains a high TTR ($> 0.10$), demonstrating its continual introduction of novel vocabulary. Thus, PubMed-derived QA exhibits greater *linguistic diversity*, whereas UniProt-based datasets provide more uniform,

**Organism Categories**

| Model | Original | Human | Bacteria | Archaea | Birds | Mammals | Vertebrates | All |
|---|---|---|---|---|---|---|---|---|
| BioMedGPT | 0.332 | 0.120 | 0.116 | 0.100 | 0.131 | 0.108 | 0.119 | 0.114 |
| Protein2Text | 0.288 | 0.362 | 0.322 | 0.283 | 0.357 | 0.354 | 0.357 | 0.337 |
| Mol-Instructions | 0.463 | 0.418 | 0.390 | 0.301 | 0.407 | 0.378 | 0.370 | 0.375 |
| ProtT3 | 0.714 | 0.013 | 0.015 | 0.013 | 0.010 | 0.017 | 0.023 | 0.016 |

**Functional Protein Categories**

| Model | Original | Kinase | Transmembrane | Structural | All |
|---|---|---|---|---|---|
| BioMedGPT | 0.332 | 0.075 | 0.133 | 0.035 | 0.065 |
| Protein2Text | 0.288 | 0.353 | 0.267 | 0.145 | 0.244 |
| Mol-Instructions | 0.463 | 0.327 | 0.200 | 0.196 | 0.247 |
| ProtT3 | 0.714 | 0.000 | 0.033 | 0.000 | 0.005 |

Table 4: **Comparing Protein LLMs on our QAProt across Organisms (top) and Functional Categories (bottom)**. We report the ROUGE-L (Lin, 2004) values, where higher values indicate better performance. **Original** indicates the reported values by the authors on their datasets. **All** refers to the evaluation across all organisms or all functional categories. Scores that are not reported by the authors are indicated with "—". We also report BLEU-2, METEOR, and LLMasJudge metrics in Appendix Tables 8, 9, and 10 respectively.

template-like phrasing; used together, they offer complementary distributions for the model training and evaluation.

### 4.3 Bias Mitigation via Inverse-frequency Sampling

Scientific literature, and by extension datasets derived from it, inherently exhibit a long-tailed distribution of topics. This can introduce significant bias, causing models to overfit to well-studied subjects (e.g., human proteins, enzyme kinetics) while underperforming on rarer, yet biologically critical, functions and organisms. While some form of this bias is desirable, for example study of human disease. But if the goal is studying rarer topics (such as rare diseases or species), the application becomes unreliable. To address this challenge and enable the creation of more robust and generalizable models, we implement a principled debiasing strategy within the QAProt pipeline.

We formalize this debiasing task as re-weighting the data distribution. First, we identify thematic topics by clustering the embeddings of source article titles. Then, to counteract the natural imbalance, we employ **inverse-frequency sampling**, where each question $i$ belonging to a topic cluster $y_i$ of size $n_{y_i}$ is assigned a sampling weight $w_i$ inversely proportional to its topic's frequency: $w_i = \frac{1}{n_{y_i}^{(\text{Topic})}}$

As shown in Figure 3a , this method effectively transforms the skewed initial distribution of topics into a more uniform one. For instance, while topics like *Mutation Effects on Function* are initially overrepresented, our sampling approach elevates the training signal for sparse topics like *Loss-of-Function Consequences*. This balanced exposure is critical for training models that can generalize across the full spectrum of protein biology, rather than simply memorizing patterns from the most common subjects. A detailed description of the sampling algorithm is available in Appendix Section E.

## 4.4 QUANTITATIVE ANALYSIS OF PROTEIN LLMS

To probe the distinct characteristics of QAProt and its utility, we evaluated several powerful, baseline sequence-to-text LLMs on our benchmark. Aggregate metrics for protein benchmarks often obscure performance variations, so we analyzed performance across taxonomic groups (e.g., archaea, mammals) or protein functional classes to better understand the generalization capabilities of models when faced with a new data distribution.

**Experimental Setup.** We evaluated four zero-shot protein LLMs: BioMedGPT, Protein2Text, Mol-Instructions, and ProtT3 . Each of model can take protein sequence as input and outputs text describing the protein or answering the query. We computed multiple performance metrics scores separately across seven organism groups and three distinct protein functional categories. Further details on model selection and inference are in Appendix B.4.

**Findings.** The results in Table 4 show a performance gap between the models' original benchmarks and their scores on QAProt. This performance gap is expected and highlights the fundamental differences between datasets derived from structured databases versus those mined from unstructured literature.

The baseline models were primarily trained on a database generated from the well-defined, structured fields of UniProt/SwissProt databases. As seen in Table 4, this results in an often template-driven and less diverse distribution of questions and answers. In contrast, QAProt sourced from PubMed manifests into a broad topical diversity of questions and answers. Due to this distribution shift, BioMedGPT BLEU-2 performance of 0.571 degraded on QAProt to as low as 0.028 for archaea and completely failed for kinases (0.000). Similarly, ProtT3's performance suggests its training data has a very different distribution, leading to a generalization gap when evaluated on QAProt benchmark. Protein2Text's performance drop on non-human proteins likely reflect the human-centric training data compared to the broader scope of QAProt.

This analysis provides further evidence that datasets derived from literature fundamentally complement those derived from structured databases. QAProt introduces a different, more diverse data distribution that can enable building truly generalizable models, and we hypothesize these powerful LLMs could benefit from training on QAProt.

| Organism | BLEU-2 | BLEU-4 | ROUGE-1 | ROUGE-2 | ROUGE-L | CIDEr |
|----------|--------|--------|---------|---------|---------|-------|
| Human    | 0.1317 | 0.0674 | 0.3092  | 0.1434  | 0.2832  | 0.8960 |
| Bacteria | 0.0986 | 0.0488 | 0.2717  | 0.1170  | 0.2479  | 0.6953 |

Table 5: Protein2Text performance after one epoch of fine-tuning on the QAProt dataset.

## 4.5 FINE-TUNING EXPERIMENT

To directly test our hypothesis that LLMs benefit from training on our literature-derived data, we conducted a fine-tuning experiment.

**Experimental Setup.** We selected the **Protein2Text** (Jararweh et al., 2025) model for this experiment due to the public availability of its pre-trained weights and fine-tuning scripts. Crucially, the original model was trained exclusively on human proteins, providing a clear test case for evaluating generalization. We fine-tuned Protein2Text on QA pairs from our cleaned QAProt dataset, which includes a wide range of human and non-human proteins. To manage computational resources, we performed efficient, adapter-only fine-tuning (keeping the core LLaMA model frozen) for a single epoch. The training process took approximately 40 hours on an 8×H100 cluster.

**Findings.** The model was evaluated on an independent test set of proteins not seen during training. Despite the limited training—just a single epoch on adapters only—the results show remarkable performance gains, as detailed in Table 5. For **human proteins**, the model's BLEU-2 score increased by 60%, from 0.082 to 0.1317. This is a substantial improvement, bringing the performance of our minimally fine-tuned model close to the 0.144 BLEU-2 score of the original, more extensively trained Protein2Text model. While for **bacterial proteins**, a domain entirely unseen by the original model, the BLEU-2 score increased by an impressive 86% (from a baseline of 0.053 to 0.0986). Thus, these improvement further demonstrates that QAProt's diverse, literature-derived data distribution fundamentally complements existing structured datasets. Even minimal exposure to QAProt provides a powerful training signal that substantially enhances a model's ability across a broader spectrum of biology.

## 4.6 ABSTRACTS VS. FULL-TEXT ANALYSIS

While abstracts are concise, full-text articles offer richer biological context, suggesting they could be a superior source for generating high-quality and diverse QA pairs. To quantify this, we conducted a pilot study comparing the outputs of our generation pipeline on 1,000 open-access articles, processing both the abstracts and their corresponding full texts.

| Metric | Full-Text | Abstracts |
|---|---|---|
| Total QA pairs | 11,912 | 7,564 |
| Total proteins | 5,304 | 1,766 |
| Unique protein names | 4,028 | 1,485 |
| Unique questions | 11,668 | 7,412 |
| Avg. question length | 51.87 | 55.80 |
| Avg. answer length | 113.63 | 88.37 |

Table 6: Comparison of QA generation from full-text articles versus abstracts.

**Findings.** Full-text articles yield nearly 60% more QA pairs and 3× greater protein diversity than abstracts, with substantially longer answers (Table 6). Despite the clear advantages of full-text data, we constructed the primary QAProt dataset from abstracts due to two key scalability challenges: (1) more than 70% articles limited public access to full texts, and (2) the high computational cost of processing longer documents at scale. This pilot study, however, underscores a valuable direction for future work, as data access and computational resources continue to improve.

## 5 CONCLUSION

The manuscript proposes QAProt, a new dataset for protein function learning. It leverages thousands of PubMed abstracts and an exceptionally powerful GPU supercomputer, delivering a QA dataset with a broader topic and linguistic diversity, which complements existing datasets. Models trained on narrow datasets perform poorly on new or underrepresented protein topics. Fine-tuning a model with QAProt data led to significant performance gains. Despite its diversity, QAProt has several limitations that provide directions for future work. First, our quality control pipeline, while effective in decreasing hallucination, is entirely LLM-based and may retain shared model biases; a formal human audit might be required to calibrate the absolute error rate. To ensure high-quality protein-to-organism mapping, we adopted a conservative filtering strategy (keep a QA only when three LLMs agree the abstract supports the specific protein–organism link; drop abstracts with ambiguous organisms or unclear entities; and discard QAs that cannot be resolved to a single organism even for homologs with shared functions), which likely discards some valid data. Second, the dataset inherits biases from the scientific literature itself, including an overrepresentation of positive findings and a topic distribution that mirrors existing research trends rather than the full breadth of protein biology. Finally, our reliance on abstracts, though necessary for scalability, is a significant limitation; expanding the pipeline to full-text articles represents a key avenue for future enhancement.

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

## A  EXTENDED DISCUSSION ON RELATED WORK

**Instruction Tuning.** Large Language Models (LLMs) have demonstrated significant capabilities in human understanding tasks, such as GPT models (Radford et al., 2019; Brown et al., 2020; OpenAI et al., 2023) and LLaMA models (Touvron et al., 2023a;b; Dubey et al., 2024). When LLMs were first introduced, they were mainly trained on next token prediction (Touvron et al., 2023a; Radford et al., 2019; Lewis et al., 2020; Liu et al., 2019; Yang et al., 2020). Instruction tuning has been proposed to align the training objective with the user objective by enhancing the model's ability to follow instructions (Zhang et al., 2024). Several models trained via instruction tuning have been proposed for a variety of tasks such as summarization (Basyal & Sanghvi, 2023), question answering (Ouyang et al., 2022; Muennighoff et al., 2023; Zheng et al., 2023), and zero-shot capabilities (Zheng et al., 2023; Ouyang et al., 2022; OpenAI et al., 2023; Wei et al., 2022; Dubey et al., 2024).

| Dataset | Sources | | | Question Format | Availability |
|---|---|---|---|---|---|
| | UniProt | PubMed | PDB | | |
| ProteinChat (Huo et al., 2024) | ✗ | ✗ | ✓ | templated | ✓ |
| UniProtQA (Luo et al., 2023) | ✓ | ✗ | ✗ | free-form | ✓ |
| PQA (Pika) (Carrami & Sharifzadeh, 2024) | ✓ | ✗ | ✗ | free-form | ✓ |
| PDB-QA (Liu et al., 2024) | ✗ | ✗ | ✓ | templated | ✗ |
| ProteinKG25 QA (Liu et al., 2024) | ✓ | ✗ | ✗ | templated | ✓ |
| Swiss-Prot QA (Liu et al., 2024) | ✓ | ✗ | ✗ | templated | ✗ |
| Mol-Instructions (Fang et al., 2024) | ✓ | ✓ | ✗ | free-form | ✓ |
| **QAProt (Ours)** | ✗ | ✓ | ✗ | free-form | ✓ |

Table 7: Comparison of major protein-QA datasets. "Sources" indicates the origin of the QAs; "Q Format" denotes question type; and "Availability" uses symbolic notation: ✓(open) and ✗(closed).

**Multimodal LLMs.** Multimodal LLMs have also been extensively applied to perform cross-modal tasks beyond the text modalities. For instance, several studies have been proposed to integrate vision and language (Liu et al., 2023a; Alayrac et al., 2022; Li et al., 2023), and integrate audio and language (Radford et al., 2022; Tjandra et al., 2017). Building on these efforts, LLMs have also witnessed prosperous adaptation to scientific and biomedical domains such as biomedical text understanding (Jararweh et al., 2024; Lee et al., 2019), biomedical QA (Wu et al., 2023; Luo et al., 2023), clinical reasoning tasks (Huang et al., 2020), and molecular structure understanding (Zhao et al., 2024; Fang et al., 2024; Cao et al., 2023; Liu et al., 2023b).

**Protein-related LLMs.** Many QA datasets have been curated to allow for instruction tuning on biomedical tasks. General biomedical QA datasets such as PubMedQA (Jin et al., 2019) and BioASQ-QA (Krithara et al., 2023) provide a wide corpus of QA examples that span diseases, genes, proteins, drugs, etc. While these datasets are comprehensive, they are limited to creating question-answer pairs from ontologies that do not capture the rich textual context present in biomedical literature. Mol-Instructions provides an extensive QA dataset specific to bio-molecules (including proteins) to extend LLMs to biomolecular structure. The sequential nature of protein primary structure lends itself to language modeling for protein characterization. For example, encoder-based LLMs trained on protein amino acid sequences have been adopted to generate a representation space that captures sequence structures (Lin et al., 2022b;a; Zhang et al., 2022; Elnaggar et al., 2021). Similarly, generative LLMs have also been proposed for a variety of protein generation tasks, such as generating 3D structure (John Jumper & Hassabis, 2021), and novel protein sequences (Madani et al., 2020; Nijkamp et al., 2022; Lv et al., 2024).

**3D Structure-based and General Biomedical QA Datasets.** Datasets like PDB-QA (Liu et al., 2024) exclusively focus on predefined queries derived from 3D protein structural information. Meanwhile, general biomedical QA datasets, such as PubMedQA (Jin et al., 2019), encompass a broad range of biomedical concepts, including diseases, proteins, and drugs, making them suitable for general biomedical QA tasks but insufficient for sequence-to-text translation tasks.

# B    EXTENDED DISCUSSION ON METHODOLOGY

## B.1    TECHNICAL IMPLEMENTATION

We implemented all scripts in Python 3.9, using libraries such as `transformers`, `torch`, `json`, `tqdm`, `multiprocessing`, and `json` for streaming large JSON files. The DeepSeek model ran on, with parallel jobs distributed across all nodes (64 GB RAM, 3200 GPUs total). To ensure QA accuracy, the DeepSeek prompt (Figure 5) restricted questions to sequence-based attributes and required answers to reflect

**Organism Categories**

| Model | Original | Human | Bacteria | Archaea | Birds | Mammals | Vertebrates | All |
|---|---|---|---|---|---|---|---|---|
| BioMedGPT | 0.571 | 0.049 | 0.040 | 0.028 | 0.050 | 0.040 | 0.043 | 0.038 |
| Protein2Text | 0.144 | 0.082 | 0.053 | 0.039 | 0.062 | 0.066 | 0.067 | 0.061 |
| Mol-Instructions | — | 0.043 | 0.030 | 0.026 | 0.029 | 0.030 | 0.030 | 0.032 |
| ProtT3 | 0.765 | 0.000 | 0.000 | 0.0001 | 0.000 | 0.000 | 0.000 | 0.000 |

**Functional Protein Categories**

| Model | Original | Kinase | Transmembrane | Structural | All |
|---|---|---|---|---|---|
| BioMedGPT | 0.571 | 0.000 | 0.110 | 0.000 | 0.038 |
| Protein2Text | 0.144 | 0.043 | 0.032 | 0.000 | 0.028 |
| Mol-Instructions | — | 0.037 | 0.000 | 0.006 | 0.014 |
| ProtT3 | 0.765 | 0.000 | 0.000 | 0.000 | 0.000 |

Table 8: **BLEU-2** scores for Protein LLMs on QAProt across (top) Organisms and (bottom) Functional Protein Categories. Higher is better.

the abstract's content. We skipped incomplete abstracts (missing `PubMed_ID`, `protein_names`, or `abstract`). During data expansion, organism matches were validated to avoid incorrect assignments, and sequence parsing cross-checked accession IDs against Swiss-Prot. We manually reviewed 100 entries, confirming consistency among QA pairs, UniProt metadata, and sequences. To assess question diversity, we counted unique questions, verifying coverage of functional, structural, and evolutionary themes.

### B.2 HALLUCINATION MITIGATION

A critical challenge in large-scale language model (LLM) generation is hallucination, where the model produces fluent but factually incorrect or unverifiable outputs. In our context, hallucinations could arise as incorrect protein mentions, incorrect organism mentions, or unrelated biological roles. We implemented multiple filtering stages to mitigate these risks:

**Keyword-Based Pruning.** We removed any QA pairs where the question or answer contained high-risk or placeholder-like terms (e.g., *"not sure"*, *"other proteins"*, *"unknown"*, *"proteinname"*, *"cells"* etc.). These often indicated vague, speculative, or non-specific answers detached from the source abstract.

**De-lexicalization of Protein Mentions.** To reduce surface-level overfitting and leakage from training, all occurrences of protein names within questions and answers were replaced with a generic placeholder (e.g., *"the protein"*). This prevents repetitions of protein names and prevents leakage of entity information into model outputs.

**Manual Verification.** In a manual audit of 100 randomly sampled entries, we found that post-filtering accuracy improved substantially, with over 90% of retained questions being specific, biologically meaningful, and clearly grounded in the abstract content.

Together, these measures greatly enhanced the reliability and usability of QAProt for training and evaluating sequence-to-text models.

**Organism Categories**

| Model | Original | Human | Bacteria | Archaea | Birds | Mammals | Vertebrates | All |
|---|---|---|---|---|---|---|---|---|
| BioMedGPT | 0.308 | 0.081 | 0.071 | 0.055 | 0.081 | 0.071 | 0.072 | 0.071 |
| Protein2Text | 0.377 | 0.248 | 0.198 | 0.160 | 0.225 | 0.219 | 0.213 | 0.209 |
| Mol-Instructions | — | 0.211 | 0.169 | 0.136 | 0.170 | 0.155 | 0.160 | 0.167 |
| ProtT3 | 0.768 | 0.004 | 0.003 | 0.006 | 0.002 | 0.004 | 0.004 | 0.004 |

**Functional Protein Categories**

| Model | Original | Kinase | Transmembrane | Structural | All |
|---|---|---|---|---|---|
| BioMedGPT | 0.308 | 0.046 | 0.066 | 0.031 | 0.042 |
| Protein2Text | 0.377 | 0.191 | 0.110 | 0.083 | 0.129 |
| Mol-Instructions | — | 0.200 | 0.071 | 0.047 | 0.110 |
| ProtT3 | 0.004 | 0.000 | 0.000 | 0.000 | 0.000 |

Table 9: Reporting **METEOR** scores to evaluate Protein LLMs on QAProt across (top) Organisms and (bottom) Functional Protein Categories. Higher values indicate better performance.

**Organism Categories**

| Model | Original | Human | Bacteria | Archaea | Birds | Mammals | Vertebrates | All |
|---|---|---|---|---|---|---|---|---|
| BioMedGPT | — | 1.30 | 1.29 | 0.97 | 1.52 | 1.31 | 1.39 | 1.30 |
| Protein2Text | — | 2.18 | 2.04 | 1.84 | 2.50 | 2.29 | 2.35 | 2.20 |
| Mol-Instructions | — | 1.73 | 1.61 | 1.02 | 2.00 | 1.59 | 1.67 | 1.60 |
| ProtT3 | — | 0.88 | 0.91 | 0.62 | 1.03 | 0.92 | 0.96 | 0.89 |

**Functional Protein Categories**

| Model | Original | Kinase | Transmembrane | Structural | All |
|---|---|---|---|---|---|
| BioMedGPT | — | 1.40 | 0.50 | 0.83 | 0.91 |
| Protein2Text | — | 4.20 | 2.00 | 1.67 | 2.62 |
| Mol-Instructions | — | 1.40 | 1.00 | 0.83 | 1.08 |
| ProtT3 | — | 1.20 | 1.00 | 0.00 | 0.73 |

Table 10: **LLM-as-Judge** average correctness scores using the DeepSeek-Chat model across (top) Organism categories and (bottom) Functional Protein categories in QAProt. Scores are measured on a 0–5 scale, where higher values indicate better judged correctness.

### B.3 EVALUATION METRICS

**BLEU Scores** (Papineni et al., 2002). The BLEU (Bilingual Evaluation Understudy) score relies on n-gram matching to calculate the performance of the generated text. The BLEU score is a precision-based metric that quantifies the number of n-grams in the generated text that are also mentioned in the ground-truth text. For example, BLEU-2 denotes bigram matching.

**ROUGE Scores** (Lin, 2004).Similarly, the ROUGE (Recall-Oriented Understudy for Gisting Evaluation) score also focuses on sequence matching. As opposed to BLEU, the ROUGE score is recall-based which calculates the amount of n-grams from the ground truth that are captured by the generated response. **ROUGE-**

**1** denotes unigram matching, **ROUGE-2** bigrams matching, and **ROUGE-L** denotes longest common subsequence matching.

**METEOR Score** (Banerjee & Lavie, 2005). METEOR (Metric for Evaluation of Translation with Explicit ORdering) weights recall and precision while performing n-gram matching. METEOR also captures high-level semantic similarity by applying stemming and synonym matching.

**CIDEr** (Vedantam et al., 2015). CIDEr (Consensus-based Image Description Evaluation) measures the similarity of a candidate caption to a set of human references by computing TF–IDF weighted n-gram overlaps. Unlike BLEU or ROUGE, it emphasizes consensus by rewarding informative words that appear consistently across human descriptions, making it better aligned with human judgments. CIDEr was validated on new datasets (PASCAL-50S, ABSTRACT-50S) with 50 captions per image and shown to outperform existing metrics in correlating with consensus-based human evaluations.

**LLMasJudge Score.** The previous evaluation metrics mostly rely on word matching between the ground truth and the prediction. Even though these metrics are usually fast, their reliance on hard matching rather than on the meaning. To this end, we utilize an LLM to act as a judge. As shown in Figure 8, the LLM will look into the ground truth and the prediction, then it will assign a score in the scale of 0-5, where 0 is the least correct( i.e., the prediction is not relevant to the ground truth) and 5 is the prediction contains all necessary information from the ground truth.

### B.4 BASELINES

**BioMedGPT** (Luo et al., 2023). BioMedGPT is a multimodal LLM that integrates molecular structures, protein sequences, and natural language text. The model aligns the three modalities to perform cross-modal tasks about proteins and molecular compounds. The model utilizes LLaMA2 (Touvron et al., 2023b) as the LLM base model. The training data was extracted from different sources, such as PubMed Central (PMC), PubChem (Kim et al., 2022), and UniProt (Consortium, 2022). We utilize the weights and default parameters released by the authors to perform inference. The inference time is 0.09 seconds per query on an 80GB H100.

**Mol-Instruction** (Fang et al., 2024). Similarly, Mol-Instruction is a multimodal LLM that integrates text, molecular compounds, and protein sequences. The model utilizes GPT3.5 to generate a QA dataset about proteins and compounds from PubMed articles. We utilize the LoRA weights published by the authors and the LLaMA-2-7b-chat-hf model from HuggingFace to perform inference. We utilize the default parameters as found in the released evaluation script. The approximate inferencing time is 18.17 seconds per query on an 80GB A100.

**ProtT3** (Liu et al., 2024). ProtT3 utilizes multimodal projection to align protein amino acid sequences and natural language text. The model is trained in two stages: protein-text retrieval and protein-text generation. During the first stage, contrastive learning objectives are utilized to extract protein features that match the description. Then, the LLM model is trained using LoRA to perform generative tasks. The authors release three different checkpoints for different tasks. We utilize the checkpoint released by the author for the QA task. The response time is 0.14 seconds per query on an 80GB H100.

**Protein2Text** (Jararweh et al., 2025). Protein2Text is a multimodal protein LLM that utilizes resampling and projection to align the text and the sequence spaces. The embeddings of the protein sequence tokens are compressed into a smaller number of tokens to preserve the context window (i.e., resampling). Next, the resampled tokens are mapped into the text space by performing the projection. We utilize the released checkpoints by the authors to perform the inference on our dataset. The response time is 0.8 seconds per query on an 80GB H100.

## C  ADDITIONAL GENERATIVE METRICS

To provide a comprehensive evaluation of the baselines in the experiment in Section 4.4, we provide more evaluation metrics to ensure fair comparison. Tables 8, 9, and 10 respectively illustrate BLEU2, METEOR, and LLMasJudge metrics on the same predictions and ground truths. The results again span the predictions across the organism and functional protein categories. "—" denotes that these values were not mentioned by the authors of the selected baselines.

## D  EXTENDED DISCUSSION ON RESOURCES

In this section, we provide a detailed account of the computational and financial resources used throughout our study. Our work relied on large language models (LLMs) both through the Oak Ridge Leadership Computing Facility (OLCF) and through the OpenRouter API, which provided access to a broad set of commercial and open-source foundation models at transparent pricing tiers. Below we describe the models, their purposes in the pipeline, and the associated costs.

### D.1  USE OF OLCF RESOURCES

We leveraged the Oak Ridge Leadership Computing Facility (OLCF) to run large-scale inference using Meta's `Llama-3.1-Instruct` model. This was primarily employed for the extraction of question–answer (QA) pairs from research abstracts. The OLCF environment allowed us to efficiently handle large-scale inference workloads that would otherwise be prohibitively expensive via commercial APIs.

### D.2  USE OF OPENROUTER

OpenRouter provided us with a unified interface to query multiple commercial models from different vendors. This service was critical for steps in our pipeline that required specialized reasoning, data cleaning, and evaluation of hallucination rates. Table 11 reports the input and output token pricing for the specific models we used, as listed on OpenRouter at the time of experimentation.

| Model | Provider | Input ($/1M tokens) | Output ($/1M tokens) | Context (tokens) |
|---|---|---|---|---|
| Llama-3.3-70B Instruct | Meta | 0.10 | 0.28 | 128K |
| Grok 3 Mini | xAI | 0.30 | 0.50 | 131K |
| DeepSeek Chat | DeepSeek | 0.56 | 1.68 | 131K |
| DeepSeek R1 | DeepSeek | 0.56 | 1.68 | 131K |
| Claude Sonnet 4 | Anthropic | 3 | 15 | 1M |
| GPT 4o | OpenAI | 5 | 15 | 128K |
| Gemini 2.5 Pro | Google | 1.25 | 5 | 128K |

Table 11: Pricing of LLMs used in this study via OpenRouter.

### D.3  LLM USAGE BY TASK

We employed LLMs for four distinct purposes:

1. **QA Extraction**: Using `Llama-3.1-Instruct` on OLCF for large-scale extraction of QA pairs from abstracts.

2. **Organism Extraction**: Using `DeepSeek-R1`, chosen for its reasoning capabilities. We ran it on around 200k different abstracts to get the main protein organism discussed.

3. **Data Cleaning**: Conducted on ∼180,000 abstracts, employing three independent evaluators for robustness: `llama-3.1-70b-instruct`, `Grok-3-Mini`, and `DeepSeek-Chat`.

4. **Data Evaluation**: To quantify hallucination rates, we evaluated 4,000 (2,000 from the original data before cleaning and 2,000 of the data after cleaning) randomly sampled abstracts across four models: `Claude Sonnet 4`, `GPT 4o`, `Gemini 2.5 pro`, and `deepseek r1`.

5. **LLMasJudge**: to produce the LLMasJudge scores, we used `DeepSeek Chat` to compare the ground truth and the prediction. We compared around 12,000 ground truths and 12,000 predictions, a total of 24,000 sentences.

## D.4 Cost Considerations

The decision to mix OLCF resources with OpenRouter was driven by cost-effectiveness. Running QA extraction on OLCF allowed us to avoid commercial charges for very large-scale inference. For tasks requiring multiple model perspectives (cleaning and evaluation), OpenRouter's pay-per-token pricing made it possible to balance accuracy with budget. Data cleaning was the most resource-intensive step at 180,000 abstracts, whereas hallucination evaluation was more limited in scope (2,000 abstracts).

Overall, this combination of public HPC resources (OLCF) and commercial APIs (OpenRouter) provided both scalability and diversity of the dataset capabilities.

---

**Algorithm 1 DebiasedSampler($D, C, m$)**: Draws a subset of size $m$ from dataset $D$ by assigning each sample a weight proportional to the inverse frequency of its class labels in columns $C$, thereby producing a class-balanced sample. Note that $m$ may be too large to satisfy the balance for a given class, given its representation in the dataset, and the user is sampling without replacement.

---

**Require:** $D$: data, $C$: list of columns to debias, $m$: desired sample size
1: $w_i \leftarrow 1.0 \; \forall i$
2: **for** $c$ in $C$ **do**                    ▷ build inverse-frequency weights
3:     counts ← ValueCounts($D[c]$)
4:     **for all** rows $i$ **do**
5:         $w_i \; \times = \; 1/\text{counts}[D_{i,c}]$
6:     **end for**
7: **end for**
8: $Z \leftarrow \sum_i w_i, \quad p_i \leftarrow w_i/Z$
9: **return** Sample($D, m, p$)

---

# E Extended Discussion on Bias Mitigation via Inverse-frequency Sampling.

**Motivation.** Large protein QA datasets often skew heavily toward well-studied topics—e.g., solubility or localization, or enzyme kinetics —while underrepresenting rare but biologically important themes. Such an imbalance can result in potential bias that can cause LLMs to overfit on common patterns and underperform on long-tail functions, undermining real-world applicability. Furthermore, certain organisms, such as humans and model organisms, are obviously well studied and are over-represented in these databases, including PubMed. This can result in model performance degrading for rare species. To mitigate such issues, we therefore added an inverse-frequency sampling feature in QAProt-seekeer, using which a user can debias training data without sacrificing overall diversity. To demonstrate this QAProt debiasing feature, we equalize topic exposure in the following experiment.

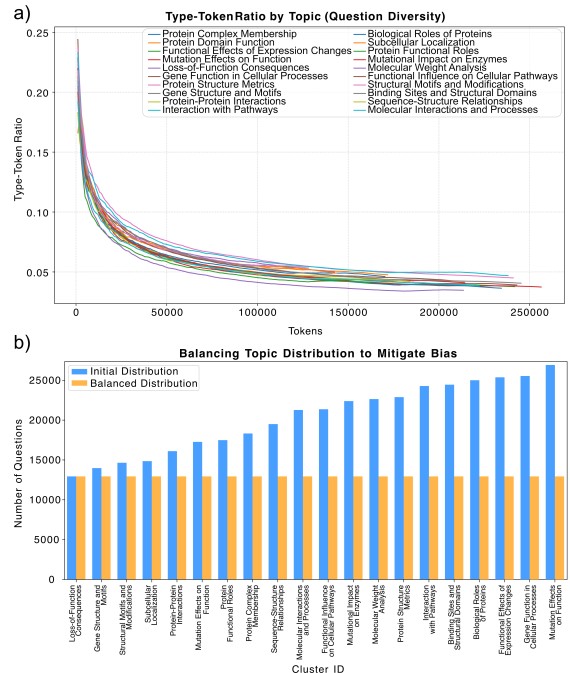

Figure 3: Topic-wise lexical diversity and debiasing of QAProt. (a) Type–Token Ratio (TTR) curves. (b) Original (blue) vs. inverse-frequency-sampled (orange) question count per topic

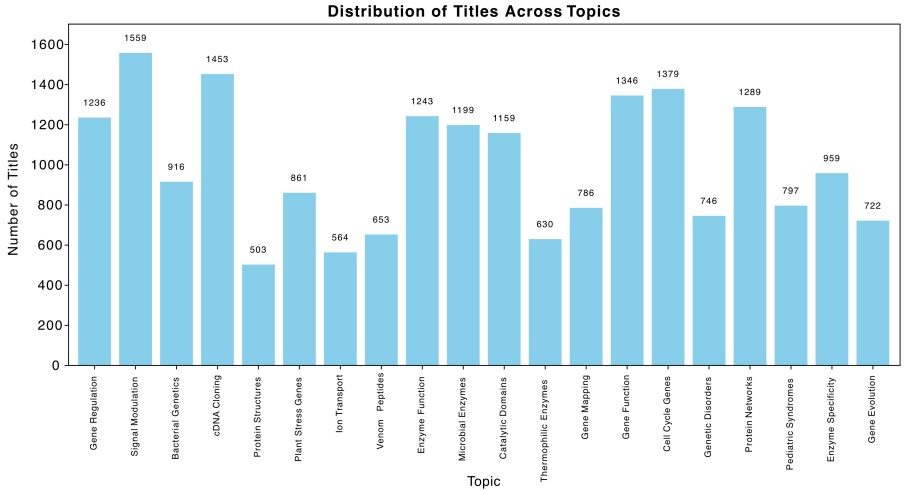

Figure 4: The distribution of 20k uniformly sampled titles across topics from our QAProt.

**Experimental Setup.**     We first embed PubMed article titles with BioMedBERT and cluster them into $K$ topic groups via K-Means. Each question inherits the cluster label(s) of its source article(s). To rebalance, we assign each question a sampling weight inversely proportional to the total number of questions in its cluster. Thus, each question $i$ receives weight

$$w_i = \frac{1}{n_{y_i}^{(\text{Topic})}} \quad \longrightarrow \quad p_i = \frac{w_i}{\sum_k w_k},$$

and we draw a balanced subset of size $N$ without replacement. Algorithm 1 describes a high-level overview implementation of this debiasing approach.

**Findings.**     Figure 3a, topics such as Mutation Effects on Function and Gene Function in Cellular Processes are disproportionately represented, while others like Loss-of-Function or Subcellular Localization are sparse. This imbalance can lead protein LLMs to underperform on underrepresented but biologically important topics. After applying inverse-frequency sampling (Figure 3b. ), question counts per cluster become less biased. This less biased distribution ensures less-represented topics receive comparable training signal, promoting more robust downstream LLM performance across the full spectrum of protein biology.

**Prompt:**
We are building an AI predictor that predicts protein attributes from the sequence. We need your help to read a research abstract and generate questions about proteins that can be predicted from the sequence and the abstract must have answers for the questions. Do not introduce yourself. Do not include greetings. Be concise, formal, and focused.

**You will be given:**

- A research abstract.

**Your task is to:**

1. For each protein name that appears in the abstract:
   - If the protein is mentioned, generate up to 10 conversations about that protein.
   - If no relevant question-answer pairs can be created, return `"conversations":  null` for that protein.
2. For each question:
   - If it mentions a specific organism/taxa/species group, list the name(s).
   - If it's about general protein properties applicable to all organisms, respond with `"All"`.
   - If unclear, respond with `"Not sure"`.

**Conversation rules:**

- Only create question-and-answer pairs that can be answered using the protein sequence alone.
- The conversation must be grounded only in the content of the abstract.
- Do not invent information. Skip speculative, vague, or hypothetical answers.
- Do not refer to the abstract or its source.

**Abstract:**
"Complexins regulate the speed and Ca(2+) sensitivity of SNARE-mediated synaptic vesicle fusion at conventional synapses. Two of the vertebrate complexins, Cplx3 and Cplx4, are specifically localized to retinal ribbon synapses. To test whether Cplx3 and Cplx4 contribute to the highly efficient transmitter release at ribbon synapses, we studied retina function and structure in Cplx3 and Cplx4 single- and double-knockout mice. Electroretinographic recordings from single and double mutants revealed a cooperative perturbing effect of Cplx3 and Cplx4 deletion on the b-wave amplitude, whereas most other detected effects in both plexiform synaptic layers were additive. Light and electron microscopic analyses uncovered a disorganized outer plexiform layer in the retinae of mice lacking Cplx3 and Cplx4, with a significant proportion of photoreceptor terminals containing spherical free-floating ribbons. These structural and functional aberrations were accompanied by behavioural deficits indicative of a vision deficit. Our results show that Cplx3 and Cplx4 are essential regulators of transmitter release at retinal ribbon synapses. Their loss leads to aberrant adjustment and fine-tuning of transmitter release at the photoreceptor ribbon synapse, alterations in transmission at bipolar cell terminals, changes in the temporal structure of synaptic processing in the inner plexiform layer of the retina and perturbed vision."

---

**Generated QA:**
**Q1:** What is the function of the protein in retinal ribbon synapses?
**A1:** The protein regulates the speed and Ca(2+) sensitivity of SNARE-mediated synaptic vesicle fusion at retinal ribbon synapses.

**Q2:** What happens when the protein is deleted in mice?
**A2:** Deletion of Cplx3 leads to a disorganized outer plexiform layer in the retina, with photoreceptor terminals containing spherical free-floating ribbons.

**Q3:** Does the protein deletion affect vision in mice?
**A3:** Yes, deletion of the protein leads to behavioural deficits indicative of a vision deficit.

**Q4:** How does the protein deletion affect synaptic transmission?
**A4:** The protein deletion leads to aberrant adjustment and fine-tuning of transmitter release at the photoreceptor ribbon synapse.

23

Figure 5: Illustrative example from our dataset showing the generation of protein-focused question-answer pairs based on a PubMed abstract (ID: 19386896).

**Prompt:**
Identify the main organism discussed in the study (i.e., the organism for which genes or proteins are investigated). Output only the organism name. If multiple organisms are mentioned, choose the one most extensively analyzed; if unclear, output "Unknown." Return only a JSON list of one organism, with no explanations or extra text.
Example outputs:
["Homo sapiens"], [""Unknown"], ...

Here is the title and abstract:
**Title:**
"Cloning and mapping of the XRN2 gene to human chromosome 20p11.1-p11.2"

**Abstract:**
"The Dhm1 gene is the mouse homologue of the dhp1(+) gene of Schizosaccharomyces pombe, which is involved in homologous recombination and RNA metabolism, such as RNA synthesis and RNA trafficking, in S. pombe. Complementation analysis showed the Dhm1 gene on a multicopy plasmid can rescue the temperature-sensitivity mutation of dhp1(ts) and the lethality of the dhp1 null mutation. This finding suggests that Dhm1 has a function in mouse similar to that of dhp1(+). The human homologue of this gene, XRN2, has been identified. A 3.6-kb transcript of XRN2 was detected in 16 tissues examined and was more abundant in testis. By radiation hybrid panel mapping, the XRN2 gene was localized to chromosome 20p11.1-p11.2 between markers D20S180 and D20S871."

**Generated Organism:** ["Homo sapiens"]

Figure 6: Illustrative example from our dataset showing the extraction of discussed organisms from a PubMed abstract (ID: 10409438).

**Prompt:**
You are an expert biomedical research assistant trained to verify the factual accuracy and relevance of question-answer (QA) pairs derived from scientific abstracts.

**You are provided with:**

- A biomedical research abstract
- A list of QA pairs referencing the abstract

**Your task is to review each QA pair and determine the following for each:**

1. **Correctness Review:**
   - Is the answer fully supported by the abstract?
   - Is the question answerable from the abstract?
   - Is the question specific and aligned with the scientific content?
   - Is the protein name in the QA correctly mentioned?
   - Does the protein name appear in the abstract?

2. **Quality Control Tag (QC):**
   - Add a new field `"QC"` to each QA that summarizes the overall review.
   - Possible values:
     - `"Pass"` – QA is fully accurate and supported.
     - `"Fail"` – QA is incorrect, misleading, or unsupported by the abstract.

Figure 7: LLM data cleaning prompt used for quality control of protein-focused QA pairs extracted from biomedical abstracts.

**Prompt:**
You are an impartial expert evaluator. Your task is to compare a predicted answer to a ground truth answer and assign a numerical score for each of the following aspects.

**Input:**

- Ground Truth: {ground_truth}
- Prediction: {predicted_answer}

**Evaluation Criteria (0 to 5):**

1. **Correctness** – Is the prediction factually and semantically accurate?
2. **Completeness** – Does the prediction include all necessary content?
3. **Conciseness** – Is the prediction free of unnecessary or redundant information?
4. **Relevance** – Does the prediction directly respond to the intended question?
5. **Fluency** – Is the prediction grammatically correct and well-structured?

Figure 8: LLM-as-Judge evaluation prompt used to score predictions against ground truth across multiple dimensions.

