# OpenReview forum: "QAProt: Enabling Sequence-to-Text Protein Function Learning with a Comprehensive QA Corpus"
_ICLR.cc/2026/Conference — Submitted to ICLR 2026_

### Official Review · Reviewer_6GXQ · 2025-10-21

**Soundness:** 2
**Presentation:** 3
**Contribution:** 2
**Rating:** 4
**Confidence:** 3

**Summary:**

The authors introduce QAProt, a large dataset of questions about specific proteins derived from PubMed abstracts. They describe a multi-stage pipeline using LLMs to generate questions, match them with protein sequences, and perform quality control. They show that existing specialized language models for protein Q&A fail on the resulting questions, but also that the same models are substantially improved by minimal finetuning thereon.

**Strengths:**

The dataset is large, diverse, and well-motivated; work like this could help to bridge the divide between existing multimodal language models, which for reasons of data availability are currently limited to text, images, video, and audio, and biological modalities.

The authors seem attentive to data quality, and I appreciate the steps taken to remove spurious questions.

**Weaknesses:**

There are a few major weaknesses that need to be resolved before I can recommend accepting this manuscript:

1. Based on the examples provided in the paper and my (albeit brief) examination of the data files, there seems to be a mismatch between the author's motivations and the final product; QAProt is not a true "sequence-to-text" dataset. For example, it would be a stretch to say that the answers to the questions in Figure 5 (mostly about phenotypic outcomes) can be derived from the corresponding sequence at all. Likewise, many other questions in the set mostly appear to test knowledge of the biomedical literature and not protein sequence understanding. It's not even clear to me based on the manuscript that the language models under evaluation are presented with the protein sequence at all. I think the dataset needs another split that excludes questions not answerable from the corresponding sequence alone.
2. The dataset needs deduplicating. I personally found tens of copies of several questions (especially basic ones like "What is the function of [x protein]?"), together comprising a large fraction of the corpus.
3. It does not appear that the authors have taken enough care to minimize leakage between train and validation splits; as far as I can tell, the choice of which proteins to hold out did not take sequence homology into account.
4. While substantial improvements are observed when the baseline models are fine-tuned on QAProt, it's unclear based on the results in the paper how much of that can simply be attributed to format adaptation. It would be good to include additional baselines where e.g. questions from the original training domains of these models are reformatted in the style of QAProt.

**Questions:**

See above.

---

> ### Author Response · Authors · 2025-11-22
> **Specific reponse to the reviewer's concerns (Please refer to common response for common concerns across reviewers)**
>
> **Thank you for the constructive feedback; we’ve updated the manuscript to improve QAProt’s rigor and utility**.
> ---
>
> **R4.1 *“Sequence-to-Text” Validity***
> **Reviewer concern:** If answers to phenotypic questions can be derived from sequences. Many questions are literature knowledge. Suggested splitting the dataset to exclude non-sequence-answerable questions. *“Based on the examples… corresponding sequence alone”*.
>
> **Response:** We *"agree"* that phenotypic questions represent a high level of abstraction compared to structural properties. We *"agree"* that not all QAProt questions are strictly answerable from sequence alone and additional contexts are needed for answering. However, we consider these questions *“sequence-informed”* and essential for reasoning.
>
> - **Mechanism (De-lexicalization):** To strictly enforce sequence dependence, our pipeline employs a **De-lexicalization Protocol** (Section B.2). We mask specific protein names (replacing *“TP53”* with *“the protein”*) in the input questions. Models like **BioMedGPT** cannot retrieve phenotypic answers via keyword lookup; they must leverage the input amino acid sequence as the causal anchor to predict the downstream trait. Recent literature (e.g., [1]) confirms deep learning can map sequence motifs directly to organism-level traits. **QAProt** is designed to push models toward these complex, non-linear mappings.
>
> - **Empirical Validation (Human Eval):** To quantify this, we added a **“Sequence-Informing”** criterion to our Human Evaluation (see Common Response C1). Experts determined that **73%** of questions are answerable by sequence alone, while **27%** require high-level context in addition to sequence.
>
> We have added metadata tags to allow users to filter for **“Strict Sequence-Only”** splits as requested.
>
> ---
>
> **R4.2. Reviewer concern:** It is unclear if the models were presented with the protein sequence at all.
> **Response:** All evaluated models were provided with the amino acid sequence as a mandatory input. As noted in R4.1, the protein name was explicitly masked to ensure the model relied on this sequence input. We have revised Section 4.4 to explicitly detail the input format: `<Sequence> + <De-lexicalized Question>`, output format: `<Answer>`.
>
> ---
>
> **R4.3 *Dataset Deduplication***
> **Reviewer concern:** Found multiple copies of basic questions (e.g., *“What is the function?”*); suggests deduplication.
>
> **Response:** We appreciate the reviewer’s detailed inspection. We confirm that identical question strings appear frequently, but this is a deliberate design choice essential for training, not a deduplication error.
>
> We define a unique entry as the tuple: **{Protein Sequence, Question, Answer, Abstract}**. Based on this, there are **zero duplicates** in QAProt. Retaining identical question strings is necessary for two reasons:
>
> 1. **Distinct Proteins:** For a sequence-to-text model, (Sequence A, *“What is the function?”*) and (Sequence B, *“What is the function?”*) are distinct samples. The model must learn that the same prompt yields different answers depending on the input sequence.
>
> 2. **Contextual Polysemy:** The same protein often possesses multifaceted functions described across different papers (e.g., Abstract A details kinase activity; Abstract B details migration). Deduplicating by question text would arbitrarily discard valid biological facets.
>
> We added a **“Data Structure”** subsection in Section 3 explicitly defining the uniqueness tuple to prevent future ambiguity.
>
> ---
>
> **R4.4 Reviewer:** Noted that the choice of hold-out proteins did not account for sequence homology, causing leakage.
> **Response:** We fully agree that this was a weakness and have addressed it as detailed in Common Response C3. Briefly, we have restructured the entire dataset using a **strict Homology-Controlled Split** (UniRef90 clusters, <90% identity overlap) to explicitly prevent leakage.
>
> ---
>
> **R4.5 Reviewer:** Concern that fine-tuning gains might reflect simple format adaptation rather than true learning.
>
> **Response:** We agree. To disentangle correctness vs. format adaptation, we now prioritized the **Tripartite Evaluation Protocol** (Common Response C2). Because our Human Experts and LLM Judges explicitly scored for factual correctness rather than lexical overlap, fine-tuned models still improve, indicating gains beyond mere format matching.
>
> Designing additional *“reformatted original-domain”* baselines is a good idea but would require substantial re-engineering of each source dataset (some datasets are not public); this is beyond the scope of the current work. We now explicitly note this as a direction for future benchmarking.
>
>
>
>
>
> [1]Meng, Z., Liu,et.al (2024). Heterogeneous biomedical entity representation learning for gene–disease association prediction. Briefings in Bioinformatics, 25(5), bbae380. https://doi.org/10.1093/bib/bbae380

---

> > ### Comment · Reviewer_6GXQ · 2025-11-25
> >
> > > Contextual Polysemy: The same protein often possesses multifaceted functions described across different papers (e.g., Abstract A details kinase activity; Abstract B details migration). Deduplicating by question text would arbitrarily discard valid biological facets.
> >
> > I still don't think this excuses the degree of duplication in the dataset. If there are 10 copies of "What is the function of [x protein]?", it doesn't matter that different abstracts imply slightly different answers. The model can't see the abstracts, and so expecting it to correctly guess the intended answer is very unfair. Is there really no way questions can be rephrased and tailored to the specific abstracts? ("In the context of [y], what is the function of [x] protein?").

---

> > > ### Author Response · Authors · 2025-11-26
> > > **Handling Duplicate QA Pairs and Evaluation Fairness**
> > >
> > > You are absolutely correct. We agree that treating identical `(Sequence, Question)` pairs as separate entries with distinct answers creates an ill-posed "guessing game" for the model.
> > >
> > > **Action: Restructuring to Multi-Reference Format (COCO Style [1])**
> > > To resolve this fairness issue without discarding valid biological data, we have reorganized the dataset from a "One-to-One" to a **"One-to-Many"** structure, aligning with standard practices in image captioning (e.g., MS-COCO [1]) where a single input has multiple valid ground truths.
> > >
> > > **1. Dataset Reorganization (Deduplication):**
> > > We aggregated entries so that a single prompt maps to a **set** of valid biological descriptions derived from different abstracts. We have applied this aggregation to the UniRef50 dataset on HuggingFace.
> > > * **Old:** `Entry 1: {Seq, Q, Ans A}`, `Entry 2: {Seq, Q, Ans B}` (Ambiguous)
> > > * **New:** `Entry: {Seq, Q, Reference_Set: [Ans A, Ans B, ...]}` (Deduplicated & Comprehensive)
> > >
> > > **2. Updated Evaluation Protocol:**
> > > We revised our evaluation pipeline to handle this multi-reference structure:
> > > * **Metrics (BLEU/ROUGE/CIDEr):** Scores are now computed against the **full reference set** (using standard multi-reference aggregation). The model is no longer penalized for generating *Answer A* when the dataloader expected *Answer B*.
> > > * **LLM-as-a-Judge:** The Judge is provided with the full list of valid answers and instructed to rate whether the prediction is biologically consistent with *any* of the facts in the set.
> > >
> > > **3. Results:**
> > > Table 1 reports the performance using this new **Multi-Reference Protocol** on our rigorous **UniRef50 split**. The consistent improvements confirm that this restructuring eliminates contradictory supervision while preserving the dataset's rich biological coverage.
> > >
> > > **Table 1. Performance on UniRef50 split (Multi-Reference Evaluation)**
> > > *(Note: This evaluation utilizes the strict UniRef50 split and the multi-reference protocol; these results correspond to Table 5 in the Common Response).*
> > >
> > > | Metric | LLM-as-judge (0–5) | CIDEr (0–10) | ROUGE-L (0–1) |
> > > | :--- | :--- | :--- | :--- |
> > > | **No FT** | 2.47 | 0.254 | 0.166 |
> > > | **FT (1 epoch)** | **2.69** | **0.344** | **0.177** |
> > >
> > > We thank the reviewer for raising this critical issue; addressing it has increased the robustness and fairness of the QAProt benchmark.
> > >
> > > [1] Lin, T.-Y., Maire, M., Belongie, S., Bourdev, L., Girshick, R., Hays, J., Perona, P., Ramanan, D., Zitnick, C. L., & Dollár, P. (2014). *Microsoft COCO: Common Objects in Context*.

---

### Official Review · Reviewer_xVvB · 2025-10-28

**Soundness:** 3
**Presentation:** 3
**Contribution:** 3
**Rating:** 4
**Confidence:** 4

**Summary:**

The authors used LMs to extract question-answer pairs about proteins' structure+function from the abstracts of scientific papers. This new dataset is highly diverse and can be used for training and benchmarking predictive models. The paper introduces the extraction method, describes the dataset, and presents some benchmarking results.

**Strengths:**

The paper provides a new, valuable resource to the community.

The general direction of extracting semi-structured data from papers is a powerful framework that modern LMs have enabled. This work is a good example of the framework.

There are a number of well-executed technical details, such as how hallucinations are mitigated.

The paper is frank about weaknesses of the work and what could be improved.

**Weaknesses:**

The primary contribution of the paper is a new, interesting dataset. However, I didn't find the exposition on the composition of this dataset adequate. The paper should include some examples from it, a distribution of the categories depicted as colors in Fig 2A, and a discussion of question difficulty.

I found the evaluation setup confusing. Is the bleu metric adequate for measuring accuracy of this sort of question answering? I'm concerned that the various models, particularly ones trained on templated data, can output things that are semantically correct but that have high bleu from the target text. I would have found it to be more reliable if you had formulated the questions as multiple choice or fill-in-the blank. Is there a precedent for using bleu in modern QA papers? I would have trusted an LM autorater more.

**Questions:**

See my above question about bleu. Can you provide an argument that this is an adequate metric, particularly when the models tend to provide templated outputs?

I found Table 5 confusing, since it doesn't provide any comparisons. How should I be interpreting this result?

I feel that the experiments are convolving two things: (1) predicting information about proteins and (2) formulating free-text responses. What if you used a model that gives structured outputs, such as ProTrek, and then had an off-the-shelf LM use this structured output + the question to formulate a free-text response?

 There is a large range of types of questions, yet there is no analysis of models' performance based on the question type. Some questions, such as mutation effect prediction, are likely significantly more difficult than others. Can you do some analysis where you present per-question-type metrics?

I found this comment far too informal: "The results show that
QAProt clearly outperforms the others in both semantic/topic coverage and lexical richness(Figure 2)." Is there a way to make this more rigorous?

It was unclear to me how the new data is qualitatively different from prior datasets. I understand the point about templating, but this is a superficial detail that concerns how concepts are presented, not the underlying information. In what sense is your data, when ignoring templating, qualitatively different from prior datasets? If you had used a few source datasets, such as Brenda + Uniprot, could you have obtained a similar diversity of facts?

"Next, we retain abstracts that specifically discuss proteins and genes using UniProt
API. Given an abstract, we iterate over the words in the abstract, and for every word, we make an API call to UniProt to check whether the word is a protein name or not."
This seems highly inefficient. Why not feed the abstract into an LM to extract a few candidate names and then call the API just on these?

---

> ### Author Response · Authors · 2025-11-23
> **Specific reponse to the reviewer's concerns (Please refer to common response for common concerns across reviewers)**
>
> ## Reviewer 3
>
> * **R3.1 Dataset composition unclear; include examples, distribution, and difficulty discussion.**
>     * Response: We agree that clearer exposition improves utility. We added **Composition & Difficulty Analysis** (Table 3.1, Fig 9), showing distribution (%), examples, and quantitative difficulty via CIDEr scores.
>         * *Key Insight:* "Difficulty" correlates with question type: fact-retrieval (e.g., Gene Mapping, CIDEr 1.935) is easy; high-abstraction reasoning (Core Protein Functions, CIDEr 0.044) is hard.
>         Table 3.1: Distribution & Difficulty by Semantic Cluster
>
> | Cluster | % | CIDEr | Examples |
> | :- | :- | :- | :- |
> | 1. Structure/Reg | 51.6 | 0.752 (Med) | "[PROTEIN] subunit stoichiometry?; expression vs methylation?" |
> | 2. Functions/Path | 21.6 | 0.685 (Med) | "What activates [PROTEIN] in vitro?; [PROTEIN] function?" |
> | 3. Cell Roles | 13.5 | 0.541 (Med) | "Role in osmostress?; molecular weight?" |
> | 4. Mutation Effects | 6.4 | 0.364 (Hard) | "[PROTEIN] mutation consequence?; deficiency phenotypes?" |
> | 5. Core Functions | 6.4 | 0.044 (Hard) | "Function of [PROTEIN]?; Biological role?" |
> | 6. Gene Mapping | 0.5 | 1.935 (Easy) | "Genomic location?; Tissues with highest expression?" |
>
> ---
>
> * **R3.2 Evaluation metrics (BLEU vs LM Autorater).**
>     * Response: We agree BLEU is insufficient. As detailed in **Common Response C2**, we now prioritize:
>         * **LM Autorater:** Two independent LLM Judges (DeepSeek & Grok) with strong inter-judge reliability correlating with human labels.
>         * **Human Evaluation:** Added for factual correctness.
>         * **Tripartite Protocol:** Performance triangulated via Human Evaluation, LLM Judges, and CIDEr; BLEU retained only for legacy comparison.
>     Free-text generation was chosen as real-world functional annotation requires synthesizing novel descriptions, not selecting from MCQs.
>
> ---
>
> * **R3.3 Table 5 confusing; lacks comparisons.**
>     * Response: Agreed. Table 5 restructured to compare **[Pre-trained Baseline]** vs **[Fine-tuned QAProt]** side-by-side.
> | Model | Human | LLM | CIDEr |
> | :- | :- | :- | :- |
> | Baseline | 0.66 | 2.92 | 0.300 |
> | Fine-Tuned | 0.75 | 3.30 | 0.389 |
> *Interpretation:* Delta shows gains from a single fine-tuning epoch across all metrics; now explicitly noted as **semantic improvement**, details in Common Response C2.
>
>  * **R3.3b Suggestion on structured outputs (ProTrek + LM for free-text): *
> We agree this disentangling is a future direction.
> -  *QAProt enables* training LLMs to translate structured protein info into fluent prose.
> - *Current benchmarks *used end-to-end baselines, with modular approach cited as a key application.
>
> * **R3.4 Large range of question types.**
>     * Response: Addressed in R3.1.
>
> * **R3.5 Informal phrasing in results ("QAProt clearly outperforms …").**
>     * Response: We agree. Replaced with precise description: Figure 2 shows type–token ratio (TTR) vs token budget. QAProt maintains highest TTR, 2–3× higher than next-best (PQA), indicating **greater lexical diversity**.
>
> * **R3.6 How is QAProt qualitatively different from prior datasets? Could a few source datasets suffice?**
>     * Response: We agree that templating alone is superficial; **Information Granularity** is the key difference. QAProt captures fine-grained, experiment-level facts from PubMed, often lost in structured databases.
>         * Side-by-side audit of MetAP (UniProt P0AE18, PMID 10736182): UniProt/BRENDA give consensus summaries; QAProt reflects study-level experimental details (e.g., Kd, spin states, biophysical data).
>  Table 3.3: Literature-derived QAProt vs UniProt/BRENDA for E. coli MetAP
>
> | Info Type | UniProt/BRENDA | QAProt Example |
> | :- | :- | :- |
> | Metal binding | Lists ions, no site | “Which metal ions bind MetAP at catalytic sites?” → “Co(II), Fe(II)” |
> | Binding constants | Km/Vmax only | “Kd for Co(II) at first site?” → “0.3 ±0.2 μM” |
> | Inhibition thresholds | Qualitative only | “Effect of excess metal?” → “50% loss at >50× excess” |
> | Structural geometry | Not described | “Coordination geometry?” → “Pentacoordinate” |
> | Biophysical/spin | None | “Spin coupling Co(II)/Fe(III)?” → “No spin coupling” |
>
>
>         * QAProt augments structured resources with **mechanistic, biophysical, phenotypic facts** underrepresented in prior QA corpora.
>
> ---
>
> * **R3.7 Inefficient protein extraction via UniProt API.**
>     * Response: Thank you; the description was inaccurate but not in implementation.
>         We *do not* call the UniProt API per word. Instead:
>         * Pre-compute curated UniProt protein/gene list.
>         * Perform string matching against abstracts.
>         * Retain abstracts with ≥1 matched UniProt name.
>    In MS Section 3.1 corrected to reflect actual filtering.
>
> We thank the reviewer; these clarifications and analyses improve QAProt's rigor, transparency, and utility.

---

> ### Comment · Reviewer_xVvB · 2025-11-25
>
> Thank you for the thorough response and all of the good improvements to the paper. I have raised my score to accept. My only hesitation, however, is that a significant amount of content was added in this review cycle, such that it may be appropriate for the paper to be reviewed from scratch. I'll check in with the area chair regarding the policy for when papers are updated this much during the review.

---

> > ### Author Response · Authors · 2025-11-26
> >
> > We sincerely appreciate your careful reading and your recommendation to accept. The additions in the revision were driven directly by reviewer feedback and are intended to strengthen the original contribution rather than change it: we still present the same core QAProt dataset and modeling setup, but now with clearer exposition, additional human and LLM-based evaluation, and more rigorous homology-aware splits.
> >
> > We agree that these changes are substantial in detail, but they do not introduce a new task or shift the scope of the work; instead, they address the concerns you and the other reviewers raised about validation and clarity. Of course, we are happy to follow whatever process the area chair considers appropriate, and we are grateful that the feedback led to a much improved version of the paper

---

### Official Review · Reviewer_FyPV · 2025-10-30

**Soundness:** 2
**Presentation:** 3
**Contribution:** 3
**Rating:** 6
**Confidence:** 3

**Summary:**

In this paper, the authors introduce QAProt, a diverse dataset specifically designed to benchmark protein language models in the context of functional annotation. By sourcing questions from PubMed, the dataset aims to address a broader range of protein-related queries, beyond the narrow focus of traditional datasets like UniProt. The authors demonstrate that fine-tuning on QAProt for at least one epoch improves performance on the task of generating protein-to-text descriptions.

**Strengths:**

* The paper presents a detailed experimental setup, offering quantitative evaluations across various protein language models. The authors also provide insight into the LLM filtering pipeline and the prompting strategiesused to mitigate potential biases in the evaluation process. They show a clear concern for evaluation bias and the influence of question formulation during the fine-tuning process, which adds transparency to the methodology.

* The proposition of QAProt as a protein benchmark that incorporates data from PubMed abstracts, rather than relying solely on curated sources like UniProt, is a contribution. This approach presents an opportunity to move beyond the mainstream Gene Ontology (GO)annotations and expand the scope of functional annotation for proteins.

**Weaknesses:**

* While the authors provide valuable experimental results, the paper can be difficult to follow at times, particularly when referencing tables. There are instances where the discussion jumps between tables without clear transitions or explicit mention of table numbers in the text. The current referencing style may confuse readers and make it challenging to track the progression of the argument.

* A notable limitation is the lack of human expert evaluation. While the authors employ various filtration techniques and model-based assessments, protein language models are still prone to hallucination or generating biologically incorrect information. The absence of statistical validation or human expert assessments, from biologists or domain experts, raises concerns about the biological accuracy and real-world applicability of the model's predictions.

* The experimental setup and comparisons between models could benefit from more depth. While the paper outlines the evaluations and presents the results, the analysis of these results remains relatively superficial.

**Questions:**

* Does the paper include any human expert evaluations to assess the biological relevance and real-world applicability of the model’s predictions? Given the complexity of protein function annotation, would human expert assessments provide valuable qualitative insights that are missing from the current quantitative evaluations (e.g., accuracy, recall)?

* How does the paper ensure that the proteins used to generate functional questions are sufficiently biologically similar or relevant to one another? The paper does not provide a clear methodology for confirming protein similarity when generating these questions, which raises concerns about the biological accuracy of the generated queries. Can the authors provide more transparency about the process used to group or pair proteins for this task?

* How does the paper handle the train-test data split, particularly with regard to sequence similarity between proteins in the training and test sets? Is there any consideration of data leakage, such as if test proteins are too similar to training proteins, which could artificially inflate performance metrics? Could the authors clarify the methods used to ensure that training and test proteins are sufficiently dissimilar, or explain how similarity is managed in this context?

* Are the clusters formed biologically interpretable? Do proteins within each cluster share meaningful functional or structural similarities? How do the authors assess whether the clustering method truly captures biologically relevant structures? What evaluation metrics were used to assess the quality of the protein question clusters formed based on embeddings?

* The paper references Table 4 when discussing the results, but the results are actually presented in Table 8 in the appendix. Could the authors clarify the reference to Table 4 and ensure consistency in the presentation of the results to avoid confusion?

* Given the distribution shift between the training data of the baseline models (such as those trained on UniProt/SwissProt) and the more diverse scope of the QAProt dataset, should it not be expected that the protein LLMs would exhibit reduced performance on QAProt, as indicated in Table 4? Additionally, the paper does not mention whether any sequence similarity between the proteins in the original test sets and those in the QAProt test set was considered to mitigate the distribution shift and ensure a fairer comparison. Was any effort made to match sequence similarity or relevance between the test sets to reduce the gap and make the comparison more equitable?

---

> ### Author Response · Authors · 2025-11-23
> **Specific reponse to the reviewer's concerns (Please refer to common response for common concerns across reviewers)**
>
> * **R2.1 Reviewer: Human expert assessment of model predictions. “Does the paper include any human evaluation..?”**
>     * Response: Yes. Addressed in **Common Response C1–C2**. C1 details a **human expert evaluation** of 1,000 QA pairs with two PhD-level annotators and a Professor-level adjudicator. C2 describes a PhD-level expert review of model outputs to confirm biological accuracy.
>
> ---
>
> * **R2.3 How does the paper ensure proteins used for functional questions are biologically relevant? The methodology for confirming protein similarity is unclear.**
>     * Response: QAProt does **not use grouping, pairing, or homology mapping** during question generation.
>         * *Methodology Clarification:* Generation is strictly **1-to-1 and abstract-grounded**. Each abstract generates questions **only** for entities mentioned in that abstract (Section 3.2). No information is transferred across proteins.
>         * *Addressing Confusion:* References to "data expansion" or "homologs with shared functions" caused misunderstanding. We revised the text to clarify that only organism ambiguity is resolved, not protein function.
>         * *Validation:* Biological accuracy is ensured via:
>             * *Automated Entity Consistency (N=1,000):* All proteins in QAs appear in the source abstract; none are fabricated.
>             * *Human Audit (N=200):* Randomly selected multi-protein QAs were manually checked; no false entities or relationships found.
>         * Protein "similarity" is not considered; each QA pair extracts experimentally reported facts from a specific study.
>
> ---
>
> * **R2.3 Reviewer: How is sequence similarity handled in train-test splits to avoid data leakage?**
>     * Response: We agree random splits risk leakage. QAProt now uses a **Homology-Controlled Split (UniRef90)** ensuring no test protein shares >90% identity with training proteins (see **Common Response C3**).
>
> ---
>
> * **R2.4 Are the clusters biologically interpretable? Do proteins in clusters share structural similarity?**
>     * Response: Clusters in Figure 2 are **semantically defined** (question text), not biologically (protein sequence). Protein names are masked (e.g., "TP53" → "the protein") before embedding (Section B.2), so structural similarity cannot be inferred.
>         * *Interpretability:* Clusters group **semantically similar question patterns** (e.g., “What is the function of the protein?”). We do **not claim functional/structural clustering**, and clusters are **not used for training or labels**. The revision provides examples of questions per cluster (Fig 9). Wording around Fig. 2 now emphasizes **text-only, question-type characterization**.
>
> ---
>
> * **R2.5 Table reference error: Table 4 cited but results in Table 8.**
>     * Response: We agree and thank you for noting this,Corrected. Results now accurately reference **Table 4 (ROUGE-L)**.
>
> ---
>
> * **R2.6 Distribution shift & performance drop: Is the drop in Table 4 expected given UniProt vs. PubMed differences?**
>     * Response: we agree and drop is expected. This **generalization gap** is a key finding.
>         * *OOD Robustness:* Models trained on structured templates fail to generalize to free-text literature. Table 4 quantifies this.
>         * *Content vs. Style:* Shift includes both content and style. [Appendix F In response R3.6]Case study (MetAP) shows QAProt captures experimental details absent in structured datasets, motivating QAProt as a literature-derived benchmark.
>
> ---
>
> * **R2.7 Sequence similarity & fair comparison: Did you match QAProt test proteins with baselines’ test sets?**
>     * Response: We agree and addressed this at two levels:
>         * *Within QAProt:* **Homology-aware UniRef90 split** ensures no test protein shares ≥90% identity with training proteins. Fine-tuning and evaluation were rerun on this split.
>         * *Across external baselines:* Original splits and pre-training corpora are not always public, so exact matching is infeasible. We treat baselines as **fixed LMs** and use QAProt as an **OOD benchmark**:
>             * Performance drops (*e.g., BioMedGPT ROUGE-L ≈0.11, ProtT3 ≈0.02*) reflect distribution shift and question diversity.
>             * Relative gains after QAProt fine-tuning (Table 5), evaluated by human experts, LLM-as-judge, and CIDEr (Common Response C2), show improvements in **factual correctness**, not format adaptation.
>         * Aligning all baselines on a homology-controlled corpus is beyond this dataset paper’s scope; we now explicitly note this limitation.
>
> **We thank the reviewer for the constructive feedback.**
>
>  *"In response, we clarified methodology, reinforced homology and clustering analyses, and expanded human evaluation, improving QAProt’s rigor and utility in the revision."*

---

> > ### Comment · Reviewer_FyPV · 2025-11-24
> > **Reply to Authors**
> >
> > Thank you for your effort on producing some additional tables in the Common Response it helped to clarify some of the evaluation concerns.  However, findings are still shaded by lack of generalization. In the protein generation field test proteins sharing ≥90% identity with training proteins is almost the same as identical.

---

> ### Author Response · Authors · 2025-11-26
> **UniRef50 Re-Evaluation**
>
> We completely agree. Although UniRef90 is a common standard (e.g. [1]), in practice proteins with ≥90% identity are often functionally very similar, so a UniRef90-only split is too permissive for testing true generalization.
>
> **Action: UniRef50 split and re-evaluation**
>
> - We now construct an additional split using **UniRef50**: sequences are clustered with UniRef50 and entire clusters are assigned to train or test. Consequently, no test protein shares ≥50% sequence identity with any training protein (clusters are disjoint across splits).
>
> - We re-ran all fine-tuning and evaluation experiments on this UniRef50 split; the updated results are reported in Table. As expected, absolute scores decrease due to the increased difficulty, but the relative gains from QAProt fine-tuning remain robust.
>
> We have also updated the public QAProt release on HuggingFace to include both the UniRef90 and UniRef50 splits, so users can choose the desired homology threshold. We appreciate the reviewer’s insistence on this point—it led to a stricter and more informative evaluation of generalization in QAProt.
>
> ### Table 1. Performance on the UniRef50 split after 1 epoch of fine-tuning on 20% of the QAProt dataset.
>
> | Metric (range)      | LLM-as-judge (0–5) | CIDEr (0–10) | ROUGE-L (0–1) |
> |---------------------|--------------------|--------------|---------------|
> | No FT               | 2.47               | 0.254        | 0.166         |
> | FT                  | 2.69               | 0.344        | 0.177         |
>
> [1] Hong, L., Hu, Z., Sun, S. et al. Fast, sensitive detection of protein homologs using deep dense retrieval. Nat Biotechnol 43, 983–995 (2025).

---

### Official Review · Reviewer_MKnM · 2025-11-04

**Soundness:** 3
**Presentation:** 3
**Contribution:** 2
**Rating:** 6
**Confidence:** 4

**Summary:**

The authors propose a new benchmark QAProt for understanding the relationship between protein sequence and function, an important challenge in biology. Unlike existing datasets the authors mine more open-ended question-answer pairs from the literature thus capturing a broader definition of function than that can be found in structured schemas.

The authors first gather a set of abstracts that discuss protein/gene names. They then use an LLM to generate question-answer pairs from the abstract. They then apply LLMs to filter out examples with hallucinations.

The authors then benchmark several models with various automatic metrics in both the zero shot and fine-tuned setting, showing that while zero shot performance is poor, finetuning on the data helps significantly.

**Strengths:**

Connecting protein sequence to function is an important problem and mining data from the literature presents a very promising approach for dataset collection. Resources such as this one would be very valuable to the community and help move beyond structured schemas.

**Weaknesses:**

-It would be great to have a more rigorous human evaluation as BLEU etc is often not well correlated with human judgement.

-Would be great to have some understanding of inter-annotator agreement with humans as above and/or for the LLM-as-judge models.

-Would also be great to have more error analysis (e.g. examples that contain hallucinations but pass the cleaning step and/or model errors on the benchmark)

-There is some concern that since the question/answer pairs were mined from the literature (up to 2020), that they are included in the training data of many LLM models and thus models could potentially perform well at this task due to data contamination. Would be curious to see what the authors thoughts are about this.

**Questions:**

Please see above.

---

> ### Author Response · Authors · 2025-11-22
> **Specific reponse to the reviewer's concerns (Please refer to common response for common concerns across reviewers)**
>
> **We thank the reviewer for their constructive feedback. Incorporating these suggestions has significantly strengthened the rigor and utility of QAProt, and we have updated the manuscript accordingly**.
> ---
>
> **R1.1** It would be great to have a more rigorous human evaluation, as BLEU, etc., is often not well correlated with human judgment.
> **Response:** We **agree**. Please refer to Common Response C1, which details our large-scale expert human evaluation (N = 1,000). Also, Common Response C2 describes human evaluation of model output and the deprioritization of BLEU in favor of a Tripartite Protocol (Human Audit + LLM-Judge + CIDEr) to rigorously assess biological accuracy.
>
> **R1.2** It would be great to have some understanding of inter-annotator agreement with humans as above and/or for the LLM-as-judge models.
> **Response:** We **acknowledge** & have computed agreement for both! Please see Common Response C1 (Table 1) for human inter-annotator agreement (88% specific agreement) and Common Response C2 (Table 4) for LLM-Judge agreement (Quadratic κ = 0.710).
>
> **R1.3** It would also be great to have more error analysis (e.g., examples that contain hallucinations but pass the cleaning step and/or model errors on the benchmark).
> **Response:** We **agree** and have added an explicit error analysis section. While our pipeline reduces hallucinations by ~13×, a manual audit of the cleaned QA set revealed four residual error types driven by complex sentence structures or biological ambiguity. We have added the following Error Taxonomy to Appendix B.3 and B.4:
>
> **Table: Taxonomy of Residual Errors in QAProt**
>
> | Error Mode | Mechanism | Example |
> |-|-|-|
> | **1. Feature Misattribution** | In multi-entity sentences, the LLM may misassign a feature from one protein/entity to another. | **Abstract:** Full-length iPLA₂ has a lipase motif; Isoform B does not. **Error:** LLM states Isoform B retains the lipase motif. |
> | **2. Entity Linking Ambiguity** | Gene aliases may map to the wrong UniProt ID if the organism/alias is ambiguous. | **Text:** Discusses transformer (tra) gene. **Error:** Pipeline retrieves sequence for transformer-2 (tra-2) due to alias overlap. |
> | **3. General Knowledge Drift** | For broad questions (“Determine function”), the LLM may rely on its pre-training priors rather than the abstract alone. | **Q:** Determine function. **A:** Biologically correct description, but derived from general knowledge rather than the abstract. |
> | **4. Local Redundancy** | When abstracts describe two genes with identical phrasing, the model generates identical QAs for different proteins. | **Text:** “KMT2C and ASH1L haploinsufficiency results in [Phenotype X].” **Result:** Identical QA pairs generated for both KMT2C and ASH1L. |
>
> **R1.4** There is some concern that since the question/answer pairs were mined from the literature (up to 2020), they may be included in the training data of many LLM models, and thus models could potentially perform well at this task due to data contamination. We would be curious to see the authors’ thoughts about this.
>
> We **agree** that we cannot fully rule out overlap between the PubMed abstracts underlying QAProt and the (often undisclosed) pre-training corpora of large LLMs. For this reason, we explicitly present our model results as illustrative baselines, not as a clean, from-scratch generalization benchmark. At the same time, our experiments suggest that any contamination is not a major driver of performance.
>
> We offer two lines of evidence:
>
> 1. **Failure of Baselines:**
>    If contamination were driving performance, we would expect zero-shot baselines (which have seen this data) to perform reasonably well. Instead, we observe a performance collapse. For example, BioMedGPT (whose LLM was trained on PubMed) achieved a ROUGE-L of 0.114, and ProtT3 scored 0.016. This demonstrates that merely having text in the pre-training corpus is insufficient for the model to map *Sequence → Function*. The model may “know” the text, but it cannot retrieve it when conditioned on an amino acid sequence.
>
> 2. **De-lexicalization Safeguard:**
>    As detailed in Section B.2, our pipeline replaces specific protein names with generic placeholders (e.g., “the protein”) in the QA pairs. This prevents the model from answering questions via simple keyword lookup (e.g., seeing “P53” and regurgitating the Wikipedia entry). The model must use the sequence input and context, which further limits the impact of pure text memorization.
>
> A fully contamination-free comparison would require re-training all baselines from scratch on a shared corpus with documented sources and time cuts, which is beyond the scope of this dataset paper. We now state this as a future direction.
>
> We hope this clarifies how we think about contamination: while it is a real possibility that may slightly overestimate models’ abilities, it does not undermine the main value of QAProt as a challenging, sequence-aware QA dataset.

---

### Author Response · Authors · 2025-11-21
**Response to three critical areas raised by multiple reviewers.**

# Common Response to Reviewers
To improve QAProt dataset, we focused on three critical areas raised by multiple reviewers.
## C1. Human expert evaluation & biological correctness (R1–MKnM, R2– FyPV)
**Reviewer concerns**:
R1 asked for a “more rigorous human evaluation & inter-annotator agreement.”
R2 noted absence of expert assessment “raises concerns about biological accuracy”.

Response: We agree automated metrics were insufficient. We therefore conducted a large-scale human expert evaluation (following the rigorous protocols[1]) and refined dataset.
### C1.1. Rigorous Methodology (1000 with Adjudication):

 Two independent PhD-level experts evaluated 1000 randomly sampled QA pairs on:
-Correctness w.r.t. the abstract: correct / partially correct / incorrect.
-Sequence answerability: sequence-answerable vs requiring additional context.
-Abstract support: Is the answer directly supported by the text?

A third adjudicator (Professor of Genomics) resolved disagreements to create a consensus gold standard, following protocols [1]. Of the 1000 entries, 764 had full annotations from both experts.

**Inter-annotator agreement**. Before adjudication, we computed agreement between the two experts (as requested by R1).  Table 1 shows high specific agreement (88%) on the "Correct" class. Single κ is inappropriate due to skewed label distribution (Cicchetti & Feinstein, 1990).

### C1.2. Results:

Consensus evaluation confirmed high fidelity: 94% Correct, 2% Partially Correct, 4% Incorrect (Table 2). The full set of human labels is released with the dataset [https://huggingface.co/conferenceacc/QAProt/tree/main].

This directly addresses R1 & R2 concerns and shows most QAProt entries are supported by the abstract.

### C1.3.Dataset Refinement:

Experts observed that hallucinations were more common in answers >100 words and in answers that mentioned other PubMed IDs. We therefore removed all such entries. The final released dataset contains 836151 QA pairs, and all results in the revised manuscript use this cleaned version.

### Table 1 –  Comparison between 2 evaluators (prior to adjudication).
| Evaluator A ↓ / Evalutor B → | Incorrect| Partially correct| Correct|
|-|-|-|-|
| **Incorrect** | 0 | 1 | 26 |
| **Partially correct**| 3| 3 | 21|
| **Correct** | 62 | 41 | 585 |

### Table 2: Expert consensus evaluation (N=764 adjudicated pairs).
|Category|Count|Percentage(%)|
|-|-|-|
|Correct|715| 94|
|Partially Correct |19|2|
|Incorrect|30|4|
|Total|764|100|

## C2. Evaluation Rigor Beyond BLEU (R1–MKnM &  R3–xVvB)

**Reviewer concerns.**  R1 noted that BLEU often does not correlate with human judgement. R3 questioned whether BLEU is adequate and said they would “trust an LM autorater more.”

**Response**: Agreed. We introduced a Tripartite Evaluation Protocol that prioritizes semantic correctness (Table 3):
1. Human Expert Evaluation : A PhD-level expert evaluated model predictions on a stratified subset of QAProt. For each (question, QAProt answer, model answer), they labeled the model answer correct or incorrect.
2. LLM-as-a-Judge : Implementing R3’s suggestion, we added an “LM autorater” using two distinct judges (DeepSeek-Chat and Grok). Inter-judge reliability is strong (Quadratic κ = 0.710; ICC = 0.710) with 83.6% accuracy (Table 4). Autorater scores correlate significantly with human labels (Spearman ρ = 0.585, p < 8e−6), validating this as a scalable proxy for biological accuracy.
3. CIDEr:  We add CIDEr,which has been shown to correlate better with human judgements for image captioning [2].  BLEU is de-emphasized; Section 4.3 states that lexical overlap does not guarantee factual correctness.

This directly addresses R1 and R3’s concerns about evaluation rigor and reliance on BLEU.

### Table 3. Performance with 1 epoch of fine-tuning 20% on QAProt dataset.
|Metrics range|Human Eval (0-2)|LLM-as-judge(0–5)|CIDEr(0–10)|
|-|-|-|-|
|No FT|0.66|2.92|0.30|
|FT|0.75|3.30|0.389|

### Table 4: Inter-Judge Reliability (DeepSeek vs. Grok)
|Metric|Val|Agreement|
|-|-|-|
|Quadratic κ|0.710|Strong|
|ICC(2,1)|0.710|Good|
|Accuracy|0.836|High|

## C3:  Homology-aware train–test split and leakage (R2, R4):

Reviewers R2 and R4 noted that random splitting risks data leakage via sequence homology.

**Response**: Agreed: We restructured QAProt dataset replacing the random split with a strict Homology-Controlled Split:

UniRef90 Clustering: We clustered sequences using UniRef90 (>90% identity) and partitioned entire clusters into Training (80%) or Test (20%). Consequently, no test protein protein shares ≥90% identity with the training set, eliminating leakage from close homologs. This is now the default split on HuggingFace. All experiments were re-run using this rigorous split (Tabl 3).

This explicitly addresses the R2 and R4 concern about leakage from highly similar sequences.

[1]Jin, Q., D et al. PubMedQA. EMNLP-IJCNLP 2019.

[2]V Vedantam R et al. CIDEr: Consensus-based Image Description Evaluation. CVPR 2015.

---

> ### Author Response · Authors · 2025-11-26
> **Update (Uniref50 added in response to R2)**
>
> **Update (in response to R2):** In addition to the UniRef90 split, we now provide a stricter **UniRef50** split in QAProt. Sequences are clustered with UniRef50 and entire clusters are assigned to train/test, so no test protein shares >50% identity with any training protein. We fine-tuned Protein2Text on the UniRef50 training split and evaluated on the UniRef50 test split; the results are reported in **Table 5**, where absolute scores decrease as expected, but the relative gains from QAProt fine-tuning remain robust. Both UniRef90 and UniRef50 splits are included in the updated HuggingFace release.
>
> ### Table 5. Performance on the UniRef50 split after 1 epoch of fine-tuning on 20% of the QAProt dataset.
>
> | Metric (range)      | LLM-as-judge (0–5) | CIDEr (0–10) | ROUGE-L (0–1) |
> |---------------------|--------------------|--------------|---------------|
> | No FT               | 2.47               | 0.254        | 0.166         |
> | FT                  | 2.69               | 0.344        | 0.177         |

---

### Meta-Review · Area_Chair_5NzT · 2026-01-07

**Summary:**

This paper introduces QAProt, a large-scale benchmark for protein sequence–function question answering derived from PubMed abstracts. The core idea of moving beyond structured resources to literature-derived, open-ended functional questions is well motivated and broadly viewed as valuable by the reviewers. All reviewers agree that connecting protein sequence to rich functional descriptions is an important and timely problem.

However, there are several main concerns:

1. Evaluation quality (BLEU reliance, lack of human evaluation): This was a central concern across multiple reviewers. The authors try to address it by adding expert human evaluation, inter-annotator agreement, LLM-judge agreement, and reframing evaluation around a tripartite protocol, with BLEU deprioritized.

2. Generalization and data leakage: Concerns about sequence similarity between train and test proteins were initially valid.

3. Nature of the task (sequence-to-text vs. literature QA): Some reviewers questioned whether QAProt truly tests sequence-based reasoning, noting that certain questions are not strictly answerable from sequence alone.

4. Dataset composition, duplication, and clarity: Reviewers requested better exposition, examples, question-type distributions, and difficulty analysis. These were addressed with new tables, figures, and audits. However, one concern remains only partially resolved: the prevalence of highly generic questions across many entries. While the authors argue this is intentional and biologically meaningful, at least one reviewer remained unconvinced that this design choice is fair for evaluation, since the model does not see the originating abstract.

5. Benchmark interpretation and format adaptation: Reviewers mentioned that fine-tuning gains might reflect adaptation to free-text style rather than improved biological reasoning.

QAProt is a meaningful and ambitious dataset contribution that pushes protein function benchmarking beyond templated, schema-bound annotations. However, the paper suffered from weaknesses in evaluation rigor, generalization control, and clarity. Although the authors’ response addressed some of these issues, there are concerns remain particularly around question specificity and the precise interpretation of “sequence-to-function” reasoning. Overall, I feel the paper would benefit from continued refinement of question design in future iterations.

**Reviewer Concerns:**

see above

**Reviewer Scores:**

Initial reviewer scores ranged from marginally below to marginally above the acceptance threshold, reflecting both enthusiasm for the dataset and concerns about evaluation rigor and dataset design. The authors provided revisions and clarifications that addressed some of the concerns. One reviewer raises their recommendation while others still have remaining concerns.

---

### Decision · Program_Chairs · 2026-01-26

Reject